# Even Faster Kernel Matrix Linear Algebra via Density Estimation

**Rikhav Shah** [1] [2]   **Sandeep Silwal** [3]   **Haike Xu** [2]

## Abstract

This paper studies the use of *kernel density estimation* (KDE) for linear algebraic tasks involving the *kernel matrix* of a collection of $n$ data points in $\mathbb{R}^d$. In particular, we improve upon the best existing algorithms for computing the following up to $(1 + \varepsilon)$ relative error for a Gaussian kernel matrix and other kernels: matrix-vector products, matrix-matrix products, the spectral norm, and sum of all entries. The runtimes of our algorithms depend linearly on the dimension $d$, sub-quadratically in the number of points $n$, and polynomially on the target error $\varepsilon$. Importantly, the dependence on $n$ in each case is far lower when accessing the kernel matrix through KDE queries as opposed to reading individual entries. Our improvements over existing best algorithms (particularly those of (Backurs et al., 2021) (ICML '21)) for these tasks reduce the polynomial dependence on $\varepsilon$, and additionally decrease the dependence on $n$ in the case of computing the sum of all entries of the kernel matrix. For example, we reduce the power of $1/\varepsilon$ from $\approx 7.7$ to $\approx 3.2$ for a $1 - \varepsilon$ relative error estimation of the spectral norm of a Gaussian kernel matrix. We complement our upper bounds with several lower bounds for related problems, which provide (conditional) quadratic time hardness results and additionally hint at the limits of KDE based approaches for the problems we study.

## 1. Introduction

Kernel functions and their associated kernel matrices have been extremely influential in practice, for example in "classical" machine learning via the so called kernel methods (Hofmann et al., 2008; Murphy, 2012; Backurs et al., 2017), as well as in "modern" machine learning, where they lie at the heart of the *attention mechanism* in transformers (Vaswani

et al., 2017; Zandieh et al., 2023; Alman & Song, 2023; Han et al., 2024; Indyk et al., 2025).

One common thread across these two eras is the *computational burden* of working with kernel matrices: given a dataset $X$ as input, initializing kernel matrices exactly requires $\Omega(n^2 d)$ time naively, and assuming standard complexity conjectures such as SETH, one cannot hope to do better than $\Omega(n^2)$ time in the exact or very precise regimes when $d = \omega(\log n)$ (Backurs et al., 2017). The quadratic bottleneck is especially salient in the case of large $n$, which is common in practice. Fortunately, SETH-based lower bounds typically only rule out extremely accurate computation (e.g. with additive error $e^{-\omega(\log n)}$), so there is still hope of reasonable *approximation* algorithms for kernel matrices, which has led to a long and fruitful line of work; see (Charikar & Siminelakis, 2017; Backurs et al., 2018; 2019; Charikar et al., 2020; Alman et al., 2020; Backurs et al., 2021; Bakshi et al., 2023; Charikar et al., 2024; Indyk et al., 2025) and references therein.

The goals of our paper are to develop much faster algorithms for approximating fundamental quantities related to kernel matrices, which are of interest to both theory and practice, including matrix-vector products, kernel sums, top eigenvector computation, and more (see Section 2). For simplicity, in the main body, we focus on the Gaussian kernel ($e^{-\|x-y\|_2^2}$) and in the appendix we also cover analogous results for other kernels such as the Laplacian or Exponential kernels.

Much of the aforementioned algorithmic work on kernel matrices has been driven by a remarkable datastructure for *kernel density estimation* (KDE): Specialized to the Gaussian case, we can take as input $n$ points $X = \{x_1, \cdots, x_n\} \subset \mathbb{R}^d$, $\varepsilon, \mu \in (0, 1)$, pre-process $X$ in $\widetilde{O}(dn/\mu^{p_g})$ time, and return a datastructure $\mathcal{D}$ such that for any query point $y \in \mathbb{R}^d$, with probability $1 - \frac{1}{\text{poly}(n)}$, we have

$$\sum_{i=1}^{n} \frac{e^{-\|y-x_i\|_2^2}}{n} \leq \mathcal{D}(y) \leq (1 + \varepsilon) \sum_{i=1}^{n} \frac{e^{-\|y-x_i\|_2^2}}{n} + \mu$$

where currently $p_g = 0.173 + o(1)$ (Charikar et al., 2020) is the best known exponent so far. A KDE query is intimately tied to the kernel matrix $K$ whose entries are given by $K_{ij} = e^{-\|x_i-x_j\|_2^2}$. Querying the KDE datastructure with $y = x_i$ approximates the $i$th row sum of $K$. Thus, many algorithmic

[1] Authors listed alphabetically [2] Massachusetts Institute of Technology [3] University of Wisconsin-Madison. Correspondence to: Sandeep Silwal <silwal@cs.wisc.edu>.

*Proceedings of the 43$^{rd}$ International Conference on Machine Learning*, Seoul, South Korea. PMLR 306, 2026. Copyright 2026 by the author(s).

works have used these queries to simulate downstream linear algebra algorithms for kernel matrices, without paying the naive $\Omega(n^2)$ time to construct them (e.g. see many of the aforementioned works).

Our algorithms take in a dataset $X$ and only operate on $K$ through KDE queries, and thus do not initialize $K$ exactly. We also do not make any structural assumptions on $K$ such as having entries bounded away from 0 (Bakshi et al., 2023) or imposing sparsity assumptions as in (Indyk et al., 2025). Since our algorithms indirectly access $K$ through KDE queries, our algorithms are modular, allowing one to plug in any KDE algorithm in theory or in practice.

## 2. Results and Comparison to Prior Best

Our results can be summarized as follows. In terms of upper bounds, we first develop a very useful primitive: we devise a faster algorithm for approximately computing non-negative kernel matrix vector products (Appendix B). We then study the application of this algorithm to two fundamental problems: multiplication of kernel matrices and computation of the spectral norm (top eigenvalue), with a witnessing vector. For matrix multiplication, we obtain an improved error estimate over previous approaches (Appendix B.2). For spectral norm estimation, we strengthen the analysis of "noisy" power iteration appearing in (Backurs et al., 2021), resulting in the optimal asymptotic relationship between the error of the top eigenvalue desired and the error of the approximate matrix-vector products required (Appendix C). We also give an improved algorithm for estimating $\mathbf{1}^\top K \mathbf{1}$, i.e. the sum of the entries of the kernel matrix (Appendix D.1).

Lastly, we present lower bounds, based on the SETH hypothesis, which show quadratic time hardness results for problems related to our upper bounds (Table 2). We also give evidence for why our non-negative matrix vector product (Appendix B.1) and kernel sum algorithms could be (near) the limit of KDE based algorithms (Appendix D.1).

Our upper and lower bound results are summarized in Table 1 and Table 2 respectively, and details follow. Note that since we are focused on theoretical improvements, we only compare to the best existing algorithms, mostly from (Backurs et al., 2021). We now discuss our results further.

### 2.1. Approximate Matrix-Vector Products

The first algorithmic problem we consider is the following: given a Gaussian kernel matrix $K$ and a vector $y$, output an approximation of the matrix-vector product $Ky$. We specialize to the case where $y$ is an entry-wise non-negative vector, as in prior work (Backurs et al., 2021).

**Definition 2.1** ($\varepsilon$-Non-negative Matrix-Vector Product, (Backurs et al., 2021)). *Given a non-negative vector $y$ and precision $\varepsilon$, return $x$ such that $x = Ky + e$ where*

$\|e\|_2 \leq \varepsilon \|Ky\|_2$ *and $e$ is entry-wise non-negative.*

The algorithm of (Backurs et al., 2021) returns $x$ as in Definition 2.1 in $\widetilde{O}\left(\frac{dn^{1+p_g+o(1)}}{\varepsilon^{3+3p_g+o(1)}}\right) = \widetilde{O}\left(\frac{dn^{1.173+o(1)}}{\varepsilon^{3.346+o(1)}}\right)$ time for the Gaussian case[1]. Our first result obtains a faster algorithm, removing more than a $1/\varepsilon$ factor in the bounds of (Backurs et al., 2021).

**Theorem 2.1** (Special case of Theorem A.1). *There is an $\varepsilon$-non-negative approximate matrix-vector product algorithm (Algorithm 1) satisfying Definition 2.1 for the Gaussian Kernel matrix which runs in time $\widetilde{O}\left(\frac{dn^{1+p_g+o(1)}}{\varepsilon^{2+p_g+o(1)}}\right) = \widetilde{O}\left(\frac{dn^{1.173+o(1)}}{\varepsilon^{2.173+o(1)}}\right)$.*

Our theorem achieves a stronger guarantee: it can approximate every coordinate of $Ky$ up to a multiplicative $1 + \varepsilon$ and additive $\varepsilon/\sqrt{n}$ factor term. See Appendix B for the proof. In Appendix B.1 we additionally give an argument, suggesting that among a natural class of algorithms which achieve our per-coordinate error, the running time of $n^{1+p_g}$ maybe necessary, where $p_g$ is the best exponent in a KDE query (currently $p_g = 0.173 + o(1)$ (Charikar et al., 2020)).

**Remark 2.1.** While our primary Matrix-Vector Product algorithm focuses on non-negative vectors, we demonstrate that relaxing this constraint to general mixed-sign vectors may encounter conditional quadratic-time hardness barriers: in Appendix E and Theorem 2.6, we show that allowing general mixed-sign vectors renders a related problem conditionally hard (requiring nearly quadratic time under SETH).

Our non-neg. matrix vector product algorithm also implies faster approximate matrix-multiplication algorithm of a Gaussian kernel matrix $K$ with any entry-wise non-neg. matrix $A$ (e.g. another kernel matrix). In time $n$ times a single non-neg. matrix vector query, we can output an approximation $B$ to $KA$ whose frobenius norm error $\|KA - B\|_F$ is bounded by $\varepsilon\|KA\|_F$, rather than $\varepsilon\|K\|_F\|A\|_F$, as is typical of approximate matrix multiplication algorithms; see Appendix B.2 for details. Our non-neg. matrix vector product is also central to the following two subsections.

### 2.2. Approximating the Top Eigenvalue

The second (and arguably our main technical focus) algorithmic problem we consider is: given a Gaussian kernel matrix $K$, output a unit vector $u$ such that $u^\top K u$ approximates the top eigenvalue $\lambda_1(K)$. Many prior works have studied sublinear approximation algorithms in more general settings, such as a symmetric or PSD matrix with bounded entries (Rokhlin et al., 2010; Musco & Musco, 2015; Bakshi et al., 2020; Swartworth & Woodruff, 2023; Bhattacharjee et al., 2024), without any kernel structure. The most relevant

---

[1] $\widetilde{O}(\cdot)$ suppresses factors of the form $\log^{O(1)}(n/\varepsilon)$.

| Problem | Our Upper Bounds | Prior Work | Improvement |
|---|---|---|---|
| Non-negative Matrix-Vector Product: 

 Compute $Ky + e$ where $\|e\|_2 \leq \varepsilon \|Ky\|_2$ | $\widetilde{O}\left(\dfrac{n^{1+p_g}}{\varepsilon^{2+p_g}}\right)$ 

 Theorem A.1 | $\widetilde{O}\left(\dfrac{n^{1+p_g}}{\varepsilon^{3+3p_g}}\right)$ | $\approx \dfrac{1}{\varepsilon^{1+p_g}} \approx \dfrac{1}{\varepsilon^{1.173}}$ |
| Top Eigenvalue: 

 Output unit vector $y$ with $y^\top Ky \geq (1-\varepsilon)\lambda_1(K)$ | $\widetilde{O}\left(\dfrac{n^{1+p_g}}{\varepsilon^{3+p_g}}\right)$ 

 Theorem A.3 | $\widetilde{O}\left(\dfrac{n^{1+p_g}}{\varepsilon^{7+4p_g}}\right)$ | $\approx \dfrac{1}{\varepsilon^{4+3p_g}} \approx \dfrac{1}{\varepsilon^{4.519}}$ |
| Non-negative Vector-Matrix-Vector Product: 

 Compute a $1 + \varepsilon$ approximation to $y^\top Ky$ | $\widetilde{O}\left(\dfrac{n^{1+p_g}}{\varepsilon^{2+p_g}}\right)$ 

 Theorem D.1 | - | - |
| Kernel Sum: 

 Compute a $1 + \varepsilon$ approximation to $\mathbf{1}^\top K \mathbf{1}$. | $\widetilde{O}\left(\dfrac{n^{\frac{1+p_g}{2}}}{\varepsilon^4}\right)$ 

 Theorem A.4 | $\widetilde{O}\left(\dfrac{n^{\frac{2+5p_g}{4+2p_g}}}{\varepsilon^{\frac{8+6p_g}{2+p_g}}}\right)$ | $\approx \dfrac{n^{0.07}}{\varepsilon^{0.16}}$ |

*Table 1.* Summary of our upper bound results for the Gaussian kernel matrix. $p_g = 0.173 + o(1)$ (Charikar et al., 2020) denotes the best known exponent for the Gaussian kernel (Charikar et al., 2020) The prior results are from (Backurs et al., 2021), which are the fastest theoretical algorithms, to the best of our knowledge. The "Improvement Factor" column highlights the asymptotic reduction in complexity provided by our algorithms. All upper bounds are also multiplied by $d$ (including those of (Backurs et al., 2021)) which we omit for simplicity. We also omit all $+o(1)$ factors in the exponents for simplicity. We refer to the specific theorems for details.

| Problem | Our Lower Bounds |
|---|---|
| Number of row/column samples required 

 to estimate $s(K)$ up to $1 + \varepsilon$ factor | $\Omega\left(\dfrac{\sqrt{n}}{\varepsilon^2}\right)$ 

 Theorem 2.5 |
| Compute $v^\top Kw$ for $v, w$ non-negative vectors 

 up to $n^{O(1)}$ factor | $\Omega(n^{2-\alpha})$ for any $\alpha > 0$ (SETH) 

 Theorem 2.7 |
| Compute $s(K) = \mathbf{1}^\top K \mathbf{1}$ for "asymmetric" kernel matrix $K$ 

 where rows and columns are indexed by different points | $\Omega(n^{2-\alpha})$ for any $\alpha > 0$ (SETH) 

 Corollary F.1 |
| Compute a $n^{O(1)}$ approximation to $\sigma_1(K)$, 

 the top singular value of an "asymmetric" kernel matrix $K$ | $\Omega(n^{2-\alpha})$ for any $\alpha > 0$ (SETH) 

 Corollary F.2 |
| Compute a $n^{O(1)}$ approximation to a non-negative 

 matrix vector product of an "asymmetric" kernel matrix $K$ | $\Omega(n^{2-\alpha})$ for any $\alpha > 0$ (SETH) 

 Corollary F.3 |

*Table 2.* Summary of our lower bounds. 1st row: (Backurs et al., 2021) gave a lower bound of $\Omega(\sqrt{n})$. See the theorems for details.

work in this line is (Swartworth & Woodruff, 2025), which shows how to achieve an *additive error* estimate of $\lambda_1(K)$. They sample a $O(1/\varepsilon) \times O(1/\varepsilon)$ principal sub-matrix of $K$ and use only those entries to compute a unit vector $u$ which satisfies $u^\top Ku \geq \lambda_1(K) - \varepsilon n$ in time $\widetilde{O}\left(1/\varepsilon^2\right)$. Without access to the underlying kernel structure, *additive error* is the best one can hope for (Swartworth & Woodruff, 2025). In particular, a *relative error* guarantee is ruled out for sub-linear algorithms: sampling $o(n^2)$ entries cannot distinguish between $K$ the identity matrix and $K$ the identity matrix

plus a symmetric pair of off-diagonal 1s, which have norms 1 and $\sqrt{2}$.

In our setting, with access to the kernel structure, *relative errors are possible*. That is, one demands the vector $u$ returned by the algorithm satisfy[2]

$$u^\top Ku \geq (1-\varepsilon)\lambda_1(K). \tag{1}$$

---

[2]This is strictly stronger when the entries of $K$ are bounded by 1 which is our case, as $\lambda_1(K) \leq n$

This guarantee was achieved by (Backurs et al., 2021) in the same setting as our paper. To the best of our knowledge, this is the only prior work that can guarantee a worst-case relative error as (1) in subquadratic time. Their idea is to first exploit the kernel structure via KDE queries to devise a fast algorithm for approximating matrix-vector products in the sense of Definition 2.1. Then, they implement the power method using an approximate matrix-vector product algorithm in place of exact matrix-vector products.

It's important to note that the desired accuracy level of the computation of $\lambda_1(K)$ and the accuracy to which the underlying approximate matrix-vector products are performed may differ. Even though we desire a $1 - \varepsilon$ multiplicative error in computing $\lambda_1(K)$, we may potentially need to use a *different* level of error $\delta$ in the approximate non-negative matrix-vector computation of Definition 2.1 when simulating the power method (e.g. we could require the error vector $e$ satisfies $\|e\|_2 \leq \delta \|Ky\|_2$ on an input $y$).

The algorithm of (Backurs et al., 2021) sets $\delta = O(\varepsilon^2)$, i.e. matrix vector products are computed *much* more accurately than the desired accuracy of the top eigenvalue estimate. Since a smaller $\delta$ requires a more expensive running time, meaning the choice of $\delta$ has a large impact in the final running time of the top eigenvector computation. Carefully adjusting some expressions in their argument shows that $\delta = O(\varepsilon^{1.5})$ suffices, though this is the limit of their argument (see Example 1). Altogether, the final runtime stated in (Backurs et al., 2021) is $\widetilde{O}\left(\frac{dn^{1+p_g+o(1)}}{\varepsilon^{7+4p_g+o(1)}}\right) = \widetilde{O}\left(\frac{dn^{1.173+o(1)}}{\varepsilon^{7.692+o(1)}}\right)$ for the Gaussian kernel, in particular, their $\varepsilon$ dependence is at least $1/\varepsilon^{7.692}$.

We give an entirely different analysis of the power method, departing from the "classic" way it is understood, and show that $\delta = O(\varepsilon)$ both suffices and is necessary (see Appendix C for a technical overview of this result). When combined with our improvement to matrix-vector products, **we remove a $\approx 1/\varepsilon^{4.519}$ factor from the prior best result**.

**Theorem 2.2** (Special case of Theorem A.3). *Let $\varepsilon \in (0, 1)$. There exists an algorithm (Algorithm 2) which returns a scalar $\lambda$ and unit vector $u$ satisfying $\lambda_1(K) \geq u^\top Ku \geq (1 - (5/8)\varepsilon)\lambda_1(K)$ and $(1 + \varepsilon/8)\lambda_1(K) \geq \lambda \geq (1 - \varepsilon/2)\lambda_1(K)$. The overall runtime of the algorithm is $\widetilde{O}\left(\frac{dn^{1+p_g+o(1)}}{\varepsilon^{3+p_g+o(1)}}\right) = \widetilde{O}\left(\frac{dn^{1.173+o(1)}}{\varepsilon^{3.173+o(1)}}\right)$.*

Ignoring log factors, our running time can be understood as running $1/\varepsilon$ iterations of noisy-power iteration using $\delta = O(\varepsilon)$ precision for a non-negative matrix-vector product, each of which takes $\approx dn^{1+p_g}/\varepsilon^{2+p_g}$ time for $p_g = .173 + o(1)$ for the Gaussian kernel. In Proposition C.3, we give a lower bound, showing that $\Omega\left(\frac{\log n}{\varepsilon}\right)$ iterations are necessary for *any* algorithm that works by post-processing

the data collected during the noisy power method (noisy matrix vector products starting from a fixed vector). We leave open the question of whether one can prove this lower bound for more sophisticated adaptive sequences of matrix-vector queries. Lastly in Appendix G, we empirically demonstrate that the scaling $\delta = O(\varepsilon)$ is also the appropriate choice in practice.

### 2.3. Vector-Matrix-Vector Products and Kernel Sum

Another type of structured matrix queries that are interesting are vector-matrix-vector queries (Rashtchian et al., 2020): given a vector $v$, output an approximation to $v^\top Kv$. We consider the case where $v$ is entry-wise non-negative (the case of having both positive and negative entries will be discussed in Section 2.4). A simple reduction shows that computing $Kv$ up to the precision of Definition 2.1 and then taking the inner product of the noisy output with $v$ gives the following result, with running time dominated by the noisy matrix vector computation.

**Theorem 2.3** (Special case of Theorem D.1). *Let $v \in \mathbb{R}^n$ be an entry-wise non-negative vector. We can compute $v^\top Kv$ for a Gaussian kernel matrix $K$ up to a $1 + \varepsilon$ multiplicative factor in time $\widetilde{O}\left(\frac{dn^{1+p_g+o(1)}}{\varepsilon^{2+p_g+o(1)}}\right) = \widetilde{O}\left(\frac{dn^{1.173+o(1)}}{\varepsilon^{2.173+o(1)}}\right)$.*

Prior to our work, one could obtain an identical approximation result but with an increase of a $1/\varepsilon$ factor in the running time via the noisy matrix-vector product algorithm of (Backurs et al., 2021). However, substantially different algorithms are needed for the special case of vector-matrix-vector products where we wish to compute $\mathbf{1}^\top K\mathbf{1} = s(K)$, the sum of all the entries in $K$. This special case has been highlighted in practice in applications such as kernel alignment (a popular measure of similarity between kernel matrices) (Cristianini et al., 2001) and in some statistics applications (Anastasiou et al., 2023). In this special case, there is hope of going beyond the $\Omega(n)$ lower bound needed to read general $v$. Indeed, (Backurs et al., 2021) gave a $o(n)$ time algorithm which approximates $s(K)$ up to a $1 + \varepsilon$. Our improved result is the following.

**Theorem 2.4** (Special case of Theorem A.4). *In time $\widetilde{O}\left(\frac{n^{\frac{1+p_g}{2}}}{\varepsilon^4}\right)$, Algorithm 3 returns a $1 + \varepsilon$ approximation to $s(K)$ with prob. 99%.*

Our exponents in $n$ and $\varepsilon$ are smaller than the bound of (Backurs et al., 2021), which obtain a running time of at least $n^{0.659}/\varepsilon^{4.159}$ for the Gaussian kernel matrix. Our algorithm of Theorem A.4 (Algorithm 3) samples $\Theta(\sqrt{n}/\varepsilon^2)$ points, improving upon (Backurs et al., 2021) which gave a lower bound of $\Omega(\sqrt{n})$.

**Theorem 2.5.** *Suppose $n \geq \Omega(1/\varepsilon^2)$. Consider an algorithm which specifies a subset $T \subseteq [n]$, receives the set of*

*points $\{x_i, i \in T\}$ and returns a $1 + \varepsilon/100$ approximation to $s(K)$, where $K$ is the underlying kernel matrix of the full set of $n$ points, with probability at least $99\%$. Then we must have $|T| \geq \Omega(\sqrt{n}/\varepsilon^2)$.*

Lastly in Appendix D.3, we give an argument suggesting why our upper bound could be the limit of a certain natural class of KDE based algorithms for computing $s(K)$.

### 2.4. Improved Lower Bounds

We complement our upper bounds with lower bounds, demonstrating the limits of algorithms for computing on kernel matrices on problems similar to those we have introduced earlier. At a high level, all of our lower bounds encode the orthogonal vectors (OV) problem (Definition 3.1) in kernel computations, which implies they must take nearly quadratic time, assuming SETH.

First we discuss the case of computing approximate matrix vector products. Clearly, there remains a big gap in our understanding of the complexity of matrix-vector products for kernel matrices: We have near linear upper bounds for the non-negative case with strong relative-error grantees (our Theorem A.1). On the other hand, virtually no non-trivial upper or lower bounds are known for the general case where the input vector can have both positive and negative entries, a question raised in prior works (Backurs et al., 2021; Indyk et al., 2025).

Our aim is to bridge this gap and provide a stronger evidence for the hardness of computing relative error estimates of $Kx$ for general $x$. This task seems out of reach of our techniques, but we show that a related intermediate problem requires $\Omega(n^2)$ time, assuming SETH. The intermediate problem is a generalization of computing $Kx$, where the kernel matrix can be "asymmetric" up to *one* vector, and we use a slightly different kernel function $e^{-\|x+y\|_2^2}$ (note the plus) for the off-diagonal entries. We also set the diagonal to all ones. More formally, we consider the problem of computing matrix-vector products for matrices $K'$, where the rows correspond to a dataset $X = \{x_1, \cdots, x_n\}$, the columns correspond to the same set $X$ *plus the origin* $0$, the main diagonal entries are all ones, and the off-diagonal entries are given by $K_{i,j} = e^{-\|x_i+x_j\|_2^2}$ Note that in the theorem below, $K' \in \mathbb{R}^{n \times n+1}$.

**Theorem 2.6.** *Consider the matrix $K'$ defined above. Any algorithm which takes as input dataset $X$, vector $x \in \mathbb{R}^{n+1}$, and returns a $y$ such that $y = K'x + e$ with $\|e\|_2 \leq n^{-0.002} \cdot \|K'x\|_2$ requires almost quadratic time, assuming SETH.*

Note that if the input vector is entry-wise non-negative, then it is easy to achieve the above guarantee in sub-quadratic time; see the discussion in Appendix E. Thus, the problem of Theorem 2.6 is a natural 'intermediate' problem where one can obtain sub-quadratic time relative error for matrix-

vector products in the case of non-negative inputs, but the same task (conditionally) requires nearly quadratic time, for vectors with mixed signs.

Our next lower bound studies a generalization of our vector matrix vector guarantee: can we compute $v^\top K w$ for a kernel matrix $K \in \mathbb{R}^{n \times n}$ where $v$ may not necessarily be equal to $w$? We show that the problem is hard, even in the case where $v$ and $w$ are assumed to be entry-wise non-negative.

**Theorem 2.7.** *For any $d = \omega(\log n)$, computing $v^\top K w$ for non-negative $v, w$ for the Gaussian kernel matrix $K \in \mathbb{R}^{n \times n}$ up to polynomial relative error requires almost quadratic time, assuming SETH.*

The underlying construction of the above theorem has broader applications in showing hardness for computations on *asymmetric* (but square) kernel matrices: where the rows and columns may not be indexed by the same point set. Such asymmetric kernel matrices are also common in practice, and we show that essentially none of our prior upper bounds are possible for the asymmetric case. In particular, we show that in the asymmetric case, computing any of the following quantities, even up to a polynomial approximation factor, solves the OV problem and thus requires nearly quadratic time for the Gaussian kernel:

- $\mathbf{1}^\top K \mathbf{1}$ (Corollary F.1),

- Top singular value (Corollary F.2),

- Computing a non-neg. mat-vec product (Corollary F.3).

## 3. Preliminaries

$X = \{x_1, \cdots, x_n\} \subset \mathbb{R}^d$ are our input data points. To avoid confusion, we let $x(i)$ denote the $i$th coordinate of a vector $x$. Throughout the paper, $\widetilde{O}(\cdot)$ hides $\text{poly}(\log(n/\varepsilon))$ terms. For a kernel matrix $K$, we let $s(K)$ denote the sum of its entries. For a subset $S \subseteq [n]$, we let $K_S$ denote the principal submatrix where we keep the rows and columns whose indices are in $S$.

**Definition 3.1** (Orthogonal Vectors (OV))**.** *Given two sets of vectors $A = \{a_1, \ldots, a_n\} \subseteq \{0, 1\}^d$ and $B = \{b_1, \ldots, b_n\} \subseteq \{0, 1\}^d$ of $n$ binary vectors, decide if there exists a pair $a \in A, b \in B$ such that $\langle a, b \rangle = 0$.*

It is known that OV requires almost quadratic time (i.e., $n^{2-o(1)}$) for any $d = \omega(\log n)$, assuming the Strong Exponential Time Hypothesis (SETH) (Williams, 2005; Abboud et al., 2014; 2015). OV is the only lower bound assumption we use and thus they are conditional on OV.

**Organization of the Rest of the Paper** Due to the page limit, we are only able to present our theorem statements, comparison to prior work, as well as high level intuition

for some of our main theorems. In the appendix, we give the full proofs of our theorems. In Section 4.1, we give a high level intuitive overview of Theorem 2.1, our improved matrix-vector product theorem, with proofs appearing in Appendix B. In Section 4.2, we do the same for Theorem 2.2, with full details in Appendix C. Section 4.3 contains an intuitive overview of Theorem 2.4 with full details in Appendix D.1 including the proofs of Theorem D.1 and Theorem 2.5. Appendix A discusses extensions of our upper bounds to other kernel functions. Appendix E and Appendix F contain proofs of our lower bound results of Table 2, and we refer to this table with pointers to specific theorems. Lastly, while the main focus of our work is on improved theoretical guarantees, we also present empirical evaluation of our power method analysis in Appendix G, where we demonstrate that our theoretical scaling of $\delta = O(\varepsilon)$ is also the correct scaling to use in practice.

## 4. Technical Overview of Upper Bounds

### 4.1. Intuition for Theorem 2.1 (Algorithm 1)

We start with a discussion of the prior algorithm of (Backurs et al., 2021). Recall we can take $p_g = 0.173 + o(1)$ for the Gaussian kernel (Charikar et al., 2020). Given an input vector $y$, the goal is to satisfy Definition 2.1. Since we know that $\|Ky\|_2^2 \geq 1$ using the non-negativity of $y$, it suffices to obtain a $1 + \varepsilon$ multiplicative approximation and $\varepsilon/\sqrt{n}$ additive error per coordinate of $Ky$ (so that the total additive error is at most $\varepsilon \leq \varepsilon\|Ky\|_2$). Thus, let's focus on the first coordinate. For the first coordinate, we want to estimate the sum $(Ky)_1 = \sum_{j=1}^n y_j k(x_1, x_j)$. The sum above looks almost like a standard KDE query of Definition A.1. However, it is weighted by the coefficients of the vector $y$. While some KDE datastructures can handle positive weights in a white-box manner (Charikar & Siminelakis, 2017), this may not be true for all KDE datastructures, so (Backurs et al., 2021) give a black-box reduction for computing the above sum given any KDE datastructure satisfying Definition A.1.

Their analysis proceeds as follows. First, note that can ignore all coordinates $y_j \leq O(\varepsilon/n^{1.5})$ since these values can only contribute value $\leq O(\varepsilon/\sqrt{n})$ error to the above sum. (Backurs et al., 2021) then proceed by bucketing the rest of the coordinates, which are all in the range $[\varepsilon/n^{1.5}, 1]$, into geometrically increasing values by a factor of $1 + \varepsilon$. Ignoring logarithmic and $d$ factors for the sake of this discussion, this results in $O(1/\varepsilon)$ total buckets.

Inside every bucket, the coordinates are approximately the same (up to $1 + \varepsilon$), so one can effectively 'factor out' the coordinates $y_j$ in the desired sum for the first coordinate of $Ky$, and use a standard KDE query. Intuitively, this bucketing allows us to treat elements with similar magnitudes as a single group, reducing the problem to a standard un-

weighted KDE query for each bucket. (Backurs et al., 2021) show that setting a *universal* value of $\mu = O(\frac{\varepsilon^2}{n})$ in the KDE query suffices for every bucket, leading to a running time of $\widetilde{O}(\frac{n^{p_g}}{\varepsilon^{2+2p_g}})$ per bucket and a total running time of $\widetilde{O}(\frac{n^{p_g}}{\varepsilon^{3+2p}})$ per coordinate, and hence an overall $\varepsilon$-precision non-negative matrix vector product in $\widetilde{O}(\frac{n^{1+p_g}}{\varepsilon^{3+2p_g}})$.

The discussion demonstrates that the extra $1/\varepsilon$ factor is from the bucketing procedure. Our first idea is to essentially get rid of the bucketing procedure (almost) entirely! To be more precise, instead of using $\Theta(\log(n/\varepsilon)/\varepsilon)$ buckets, we only use $\Theta(\log(n/\varepsilon))$ many buckets by partitioning the range $[\varepsilon/n^{1.5}, 1]$ by endpoints that differ by a power of 2 rather than $1 + \varepsilon$. This presents one challenge: we no longer have the guarantee that inside every bucket, the coordinates are all the same up to a $1 + \varepsilon$ factor. This is what allows (Backurs et al., 2021) to reduce the summation inside a bucket (note the summation is weighted by the coordinates of the input vector $y$) into a *single* KDE query by factoring out $1 + \varepsilon$. Our second idea is instead to show that a (positively) weighted KDE sum, e.g. for $(Ky)_1 = \sum_{j=1}^n y_j e^{-\|x_1 - x_j\|_2^2}$ can always be *directly* be reduced to a single KDE query (Lemma B.1). However, an astute reader may wonder why then do we need to perform the bucketing at all? In other words, we can simply just view every entry of $Ky$ as a weighted KDE sum and reduce the problem to $n$ KDE queries. The challenges lies in the fact that if we naively do this, we need to set the additive error parameter to be smaller than $1/n^{1.5}$, which gives a much worse $n$ dependency!

Thus, our third idea is the following. Rather than using one KDE data-structure per bucket (with fixed $\mu$ as done in (Backurs et al., 2021)), we use an *adaptive* choice of $\mu$, depending on the approximate value of the bucket. We have to handle two regimes: the first is when the bucket has mass more than $1/\sqrt{n}$, i.e., consider coordinates with value $\Theta(2^t/\sqrt{n})$ for some parameter $t \geq 0$. Such buckets can only have at most $O(n/2^{2t})$ many coordinates, since the total $\ell_2$ squared sum of the input $y$ is bounded by 1. For this bucket, we scale the corresponding coordinates in $y$ by a fixed factor of $\Theta(\sqrt{n}/2^t)$ to ensure that all the coordinates are now in the range $(0, 1)$. We then evaluate the weighted KDE sum via one underlying KDE query, using our new transformed weights, using the previous idea. If we set the error for this bucket to be $\mu_t$, then the total additive error we get for the contribution of coordinates in this bucket is $\Theta(2^t/\sqrt{n})$, coming from factoring out the normalization that we performed on the coordinates $y_j$ in our weighted KDE sum, times $O(n/2^{2t})$, coming from the size of the bucket since the KDE query gives a scaled sum, times $\mu_t$. Thus, the additive error per bucket can be approximately bounded by $\approx \frac{2^t}{\sqrt{n}} \times \frac{n}{2^{2t}} \times \mu_t$. The right choice is to set $\mu \approx \frac{2^t \varepsilon}{n}$, bounding the overall additive error per bucket

by $\widetilde{O}(\varepsilon/\sqrt{n})$, which allows us to control the total error across all buckets. Buckets that are smaller than $1/\sqrt{n}$, say $\Theta(\frac{1}{2^t\sqrt{n}})$ for $t \geq 1$ are also shown to require $\mu \approx \frac{2^t\varepsilon}{n}$, although the analysis is slightly different. Thus, the overall query time is approximately

$$\frac{1}{\varepsilon^2}\sum_{t\geq 0}\frac{1}{\mu_t^{p_g}} = \sum_{t\geq 0}\left(\frac{n}{2^t\varepsilon}\right)^{p_g} = O\left(\frac{n^{p_g}}{\varepsilon^{2+p_g}}\right)$$

across *all* buckets for a fixed coordinate, avoiding the $1/\varepsilon$ overhead. One important point to note is that we have to be careful and bound the overall running time of creating all the different KDE datastructures as well. Note that Theorem A.1 only holds for non-negative vectors. This is not a limitation of our algorithm or our analysis, but could come from a fundamental barrier based on SETH, which we quantify in Theorem 2.6 and Appendix E by studying a related family of matrices where obtaining relative error for non-negative matrix vector is possible in sub-quadratic time, but is not possible with mixed signs under SETH.

## 4.2. Intuition for Theorem 2.2 ( Algorithm 2)

In our algorithm, exact matrix-vector products in the usual power method are replaced by a approximate matrix-vector products with error satisfying Definition 2.1. This method, stated precisely in Algorithm 2, produces a sequence of unit vectors $z_0, z_1, z_2, \ldots$ satisfying the following definition.

**Definition 4.1** ($\delta$-approximate power iteration). *Let $z_t, \widetilde{z}_t$ be the sequence of normalized and unnormalized unit vectors produced by $\delta$-approximate power iteration starting with $\langle v_1, z_0\rangle \geq 1/\sqrt{n}$. That is,*

$$\widetilde{z}_{t+1} = Kz_t + \|Kz_t\|\,\delta u_t \quad \& \quad z_{t+1} = \frac{\widetilde{z}_{t+1}}{\|\widetilde{z}_{t+1}\|} \quad (2)$$

*for some unit vectors $u_t$ satisfying $\langle v_1, u_t\rangle \geq 0$.*

(Note that Definition 2.1 guarantees the returned answer is entry-wise non-negative and the top eigenvector of our kernel matrices is also entry-wise non-negative via the Perron-Frobenious theorem). In this definition, $\delta$ denotes the error to which each matrix-vector product is performed. When $\delta = 0$, this is the exact power method and $z_t$ approaches span $\{v_j : \lambda_j = \lambda_1\}$. For $\delta > 0$, the best available analysis in the literature is due to (Backurs et al., 2021). They show that if $\delta = O(\varepsilon^2)$, then the power method produces a unit vector $u$ satisfying the relative error guarantee, $u^\top Ku \geq (1-\varepsilon)\lambda_1$, in $O(\log(n/\varepsilon)/\varepsilon)$ iterations. The smaller $\delta$ is, the more expensive the approximate matrix-vector products become. It's therefore desirable to improve upon the condition $\delta = O(\varepsilon^2)$. The classical analysis of the power method as well as the analysis of (Backurs et al., 2021) expresses each iterate $z_t$ in the eigenbasis of $K$ and considers the mass this vector places on the top eigenvector and the eigenvalues above versus below the threshold

$(1-\varepsilon)\lambda_1$. The analysis then shows that the mass placed below $(1-\varepsilon)\lambda_1$ is driven to 0. By adjusting some of the expressions in (Backurs et al., 2021), one can improve $\delta = O(\varepsilon^2)$ to $\delta = O(\varepsilon^{1.5})$, but the argument fundamentally breaks down for $\delta = \Omega(\varepsilon^{1.5})$ as demonstrated by Example 1.

Rather than track the mass that each iterate $z_t$ places on eigenvalues above and below $(1-\varepsilon)\lambda_1$, **our analysis tracks the mass placed on just the top eigenvector $v_1$, that is, $\langle v_1, z_t\rangle$, and the norm of the vector $\|Kz_t\|$.** This can be thought of as a kind of weighted sum of the components $\langle v_j, z_t\rangle$ in the eigenbasis where the components corresponding to small eigenvalues are weighted less. Notably, we do not make a hard cutoff between eigenvalues that are above or below $(1-\varepsilon)\lambda_1$. We directly show that $\|Kz_t\|$ is driven to $\lambda_1$ *without* appealing to an argument that shows the components in the direction of small eigenvectors become small. Our tight analysis of how MVP error affects $\lambda_1$ relative error in the power method may also be of independent interest; see Appendix C.

**Example 1** (Limitation of previous argument for $\delta = \Omega(\varepsilon^{1.5})$). Suppose one runs $\delta$-approximate power iteration for $\delta \geq (2\varepsilon)^{1.5}/(1-5\varepsilon^2)$ on a kernel matrix $K \in \mathbb{R}^{3\times 3}$ with the goal of achieving the relative error guarantee (1) in any number iterations. Suppose $K = VDV^\top$ has the eigenvalues $\lambda_1 > 0$, $\lambda_2 = (1-2\varepsilon)\lambda_1$, $\lambda_3 = 0$. Let $x_t = V^\top z_t$, $\widetilde{x}_t = V^\top\widetilde{z}_t$. This example shows that one *cannot* conclude convergence by only considering the quantities $x_t(1)$ and $\|x_t(2:3)\| = \sqrt{x_t(2)^2 + x_t(3)^2}$. In particular, the set $S = \left\{x \in \mathbb{R}^3 : x_t(1) = \sqrt{1-2\varepsilon}, \|x_t(2:3)\| = \sqrt{2\varepsilon}\right\}$ both contains vectors that do not witness $\lambda_1$, and vectors that are fixed by the noisy power method. Consider $x_t = \begin{bmatrix}\sqrt{1-2\varepsilon} & \sqrt{2\varepsilon} & 0\end{bmatrix}^\top \in S$. Then it is possible for the next iterate $\widetilde{x}_{t+1}$ to be equal to

$$\begin{bmatrix}\lambda_1\sqrt{1-2\varepsilon} & \lambda_2\sqrt{2\varepsilon} + \delta\sqrt{\lambda_1^2(1-2\varepsilon) + \lambda_2^2\cdot 2\varepsilon} & 0\end{bmatrix}^\top$$
$$= \lambda_1\begin{bmatrix}\sqrt{1-2\varepsilon} & \sqrt{2\varepsilon} - (2\varepsilon)^{1.5} + \delta\sqrt{1-8\varepsilon^2 + 8\varepsilon^3} & 0\end{bmatrix}^\top.$$

If $\delta \geq \frac{(2\varepsilon)^{1.5}}{\sqrt{1-8\varepsilon^2+8\varepsilon^3}}$, then it's possible for $x_{t+1} = x_t$, meaning that the power method makes no additional progress. On the other hand, consider $x_t = \begin{bmatrix}\sqrt{1-2\varepsilon} & 0 & \sqrt{2\varepsilon}\end{bmatrix}^\top \in S$. The quadratic form $z_t^\top Kz_t = x_t(1)^2\lambda_1 = (1-2\varepsilon)\lambda_1$, does not give an $\varepsilon$ relative error approximation of $\lambda_1$.

## 4.3. Intuition for Theorem 2.4 (Algorithm 3)

The algorithm (and proof) proceeds in three steps. Here $p_g = 0.173 + o(1)$ in the Gaussian kernel.
**Step 1:** In the first step, we show that a random $m \times m$ principal submatrix $K_A$ for $m = \Theta(\sqrt{n}/\varepsilon^2)$ preserves the kernel sum. More precisely, we show via a concentration bound calculation in Lemma D.2 that it suffices to compute the sum of the *off-diagonal* entries of $K_A$, denoted

by $s_o(K_A)$, up to additive poly($1/\varepsilon$) to get a $1 + \varepsilon$ multiplicative approximation of $\mathbf{1}^\top K \mathbf{1}$. We also prove this is the optimal sampling complexity; see Theorem 2.5, which proves a stronger version of the idea of (Backurs et al., 2021) who only sampled a $\approx \sqrt{n} \times \sqrt{n}$ submatrix.

**Step 2:** Now we approximate the off-diagonal sum of $K_A$. A natural strategy is to just initiate $m$ KDE queries with an appropriate additive error. It turns out that one needs to set the additive error $\mu$ quite low, leading to an overall $n^{1/2+p_g}$ running time (note that we are aiming for $n^{1/2+p_g/2}$). Instead, we use the high-level idea of (Backurs et al., 2021) in obtaining their final bound: we filter out all "heavy" rows of $K_A$ using KDE queries with a relatively large $\mu$ (which is faster than the prior strategy). Then for the remaining "light" rows, we know their individual contribution to the off-diagonal sum is bounded by $\approx \mu\sqrt{n}$. This means that instead of computing their contribution via KDE queries, one can further *subsample* light rows to determine their total contribution to the off-diagonal sum of $K_A$. Optimizing this, (Backurs et al., 2021) prove an overall running time of $n^{\frac{2+5p_g}{4+2p_g}}$ where $\frac{2+5p_g}{4+2p_g} = \frac{1}{2} + \frac{2p_g}{p_g+2}$. Since $p_g$ is a relatively small constant, the improvement over $1/2 + p_g$ is relatively minor.

**Step 3:** This is the main technical step that is substantially different than (Backurs et al., 2021). Once we subsample light rows, we have no choice but to either compute their sum *exactly* (which is what (Backurs et al., 2021) opt to do). We could use KDE queries since the sample size can be shown to be $\ll m$, which actually would lead to our improved running time. However, the issue here is that *creating* the KDE datastructure is too costly now since there are still $m$ vectors in the KDE sum (corresponding to the columns since only the rows have been sub-sampled). Intuitively, what is happening is that KDE queries don't really outweigh the benefit of exact sum unless one makes $\Omega(m)$ queries, where $m$ is the total number of vectors in the KDE datastructure. This does not appear to be the case here since there are relatively very few light rows, meaning few queries, (after sub-sampling) compared to the number of columns.

Thus, we employ a very different sampling strategy. Once we filter out the heavy rows, we note that it also filters out heavy *columns* since the matrix $K_A$ is symmetric. Then we sample a square *principal submatrix* of $K_A$, rather than just sampling rows as done in (Backurs et al., 2021). Thankfully, the same concentration inequality (Lemma D.2) for Step 1 can be used to analyze this sampling as well. The advantage of our sampling is now the number of rows/columns of the sub-sampled matrix, which recall we sub-sampled from only light rows and columns, is *balanced*, meaning this is exactly the regime where KDE queries are useful (the number of KDE queries will be approximately equal to the total umber of vectors in the datastructure)! Thus, we

initialize an appropriate KDE datastructure on the columns of this sampled square matrix and query all the rows to compute a sufficiently fine-grained contribution of the light rows to $s_o(K_A)$. By carefully balancing parameters across the three steps, we are able to prove a final running time of $n^{1/2+p_g/2}$, which we additionally argue in Appendix D.1 is a "natural" limit of KDE based approaches.

## 5. Conclusion and Open Questions

Our work makes significant theoretical progress on natural algorithmic problems on kernel matrices. We believe the following are the most interesting questions left open.

**Question 5.1.** *What is the smallest $\varepsilon \in (0, 1)$ such that $s(K)$ can be computed up to relative error $1 \pm \varepsilon$ in subquadratic time? In particular, is $\varepsilon < \frac{1}{n}$ possible in subquadratic time?*

We conjecture the answer to the above question to be no. Note that it is possible to prove SETH based lower bounds for extremely high precision (computing $s(K)$ up to $1 + e^{-\omega(\log n)}$ error) (Backurs et al., 2017)), but it seems new lower bound assumptions maybe needed to prove hardness for "moderately large" $\varepsilon$. The second question is a natural follow up to our non-negative matrix vector product upper bound of Theorem A.1.

**Question 5.2.** *Can we prove a quadratic time hardness for computing a matrix vector product $Kx$ for a general $x$ (i.e. not necessarily non-negative) with guarantees similar to Theorem A.1?*

We conjecture the answer is yes based on the hardness of our intermediate problem given in Theorem 2.6. A related question concerns approximating the eigenvalues of $K$ beyond the top eigenvector. Algorithmically, a bottleneck in obtaining an upper bound by using something subspace iteration or Arnoldi iteration is those algorithms require general matrix-vector queries. This was also an open question given by (Backurs et al., 2021).

**Question 5.3.** *What is the complexity of computing $\lambda_2(K)$? In particular, can we compute a factor of 2 approximation of $\lambda_2(K)$ in sub-quadratic time?*

What specifically makes this problem difficult is that we do not have control over the inner product between the error vector of computing matrix-vector products and the second eigenvector. Our Theorem A.3 strongly used that both the error vector and top eigenvector had a positive dot product; Example 2 showed that the method fails without this assumption. Our last question concerns the effectiveness of the power method for just the top eigenvalue.

**Question 5.4.** *How many $\delta$-approximate non-negative matrix-vector queries are required to estimate $\lambda_1(K)$ up to $\varepsilon$ relative error?*

Our lower bound Proposition C.3 only applies to the sequence of queries made by noisy power iteration, and only says something non-trivial when $(\delta/\varepsilon)\sqrt{n} \gg 1$. What about other (possibly adaptive) queries? In particular, the two matrices described in Proposition C.3 that are not distinguished by the power method *are* distinguished by the single query $K\mathbf{1} + (\text{noise}) - \mathbf{1}$. When $\delta = 0$ (when the matrix vector products are *exact*), it's clear one can use Krylov subspace / Chebyshev polynomials to reach $\widetilde{O}(1/\sqrt{\varepsilon})$ iterations (Musco & Musco, 2015). Can one obtain suitable control over the error when $\delta > 0$ to extrapolate this bound?

## Impact Statement

This paper presents theoretical work whose goal is to advance the field of Machine Learning. There are many potential societal consequences of our work, none which we feel must be specifically highlighted here.

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

## A. Extension of Our Upper Bound Results to Other Kernel Functions

Our main upper bound results also hold for other kernels beyond the Gaussian kernel. In this section, we survey our general results before proving them in the subsequent sections. First, we state a general definition of kernel density estimation (KDE), which generalizes the functionality to other kernels.

**Definition A.1** (Fast KDE). *A kernel function $k : \mathbb{R}^d \times \mathbb{R}^d \to \mathbb{R}$ admits a 'Fast KDE' algorithm, if given a set of $n$ points $X = \{x_1, \cdots, x_n\} \subset \mathbb{R}^d$, $\varepsilon, \mu \in (0,1)$, there exists a $p \in (0,1)$ (depending on $k$) such that we can process $X$ in $\widetilde{O}(dn/\mu^p)$ time and return a datastructure $\mathcal{D}$ such that for any query point $y \in \mathbb{R}^d$, with probability $1 - \frac{1}{\text{poly}(n)}$, we have*

$$\frac{1}{n} \sum_{i=1}^{n} k(y, x_i) \leq \mathcal{D}_X(y) \leq \frac{1+\varepsilon}{n} \sum_{i=1}^{n} k(y, x_i) + \mu. \tag{3}$$

*The query time of $\mathcal{D}_X$ is $O(\frac{d \log n}{\varepsilon^2 \mu^p})$.*

As a baseline, it can be easily checked that random sampling can guarantee the above definition for $p = 1$ and a substantial body of research has been done to develop algorithms to obtain $p < 1$ for many popular kernel functions such as the Gaussian and Exponential kernel in high dimensions; this is summarized in Table 3 below.

| Kernel | $k(x,y)$ | $p$ | Reference |
|---|---|---|---|
| Gaussian | $e^{-\|x-y\|_2^2}$ | $0.173 + o(1)$ | (Charikar et al., 2020) |
| Exponential | $e^{-\|x-y\|_2}$ | $0.1 + o(1)$ | (Charikar et al., 2020) |
| Laplacian | $e^{-\|x-y\|_1}$ | $0.5$ | (Backurs et al., 2019) |
| Rational Quadratic | $\frac{1}{(1+\|x-y\|_2^2)^\beta}$ | $0$ | (Backurs et al., 2018) |

*Table 3.* Results for kernel density estimation queries that can be used in Definition A.1. Note that $p < 1$ in all cases. The stated result in the last row assumes $\beta \geq 0$ and $\beta = O(1)$. We note that there is an alternate data structure for the rational quadratic kernel given in (Charikar et al., 2024) which achieves query time $O(\frac{\text{poly}(\log(n/\varepsilon) \cdot d)}{\varepsilon})$, i.e. it has a $\varepsilon^{-1}$ dependency at the cost of a larger polynomial dependence on $d$.

**Remark A.1** (Remark on KDE guarantees). We note that usually, the KDE guarantees are written in a slightly alternate form (Charikar et al., 2020; Backurs et al., 2021): Given a lower bound $\mu$, the datastructure returns a $1 + \varepsilon$ approximation to the sum $\frac{1}{n} \sum_{i=1}^{n} k(y, x_i)$ assuming it is at least $\mu$. The datastructure also reports the case where $\mathcal{D}_X(y) < \mu$. By adding $+\mu$ to the output in the second case, we can also obtain the weaker guarantee stated in (3), which will be convenient in our analysis.

Given the setup, we can now state our general results for other kernels, assuming access to a suitable KDE datastructure. Thus our theorems depend on $p$, which is a kernel dependent quantity (see Table 3).

Our first result concerns non-negative matrix-vector products. The proof of the theorem appears in Appendix B.

**Theorem A.1.** *Let $k$ be the Gaussian ($e^{-\|x-y\|_2^2}$) or the Laplacian ($e^{-\|x-y\|_1}$) kernels and let $p$ be the corresponding exponent in the KDE datastructure stated in Table 3 ($p = 0.173 + o(1)$ and $p = 0.5$ respectively). Let $K \in \mathbb{R}^{n \times n}$ be the associated kernel matrix for $n$ points in $d$ dimensions. There is an $\varepsilon$-non-negative approximate matrix-vector product algorithm (Algorithm 1) satisfying Definition 2.1 for $K$ running in time $\widetilde{O}\left(\frac{dn^{1+p}}{\varepsilon^{2+p}}\right)$.*

In contrast, the algorithm of (Backurs et al., 2021) returns such a $x$ in Definition 2.1 in $\widetilde{O}\left(\frac{dn^{1+p}}{\varepsilon^{3+2p}}\right)$ time[3]. Thus our result removes a $1/\varepsilon^{1+p}$ factor in the bounds of (Backurs et al., 2021) for the Gaussian and Laplacian case. Our improved algorithm (Algorithm 1) for Theorem Theorem A.1 is only valid for these two kernels. However, we note that the $\widetilde{O}\left(\frac{dn^{1+p}}{\varepsilon^{3+2p}}\right)$ time algorithm of (Backurs et al., 2021) is also valid for the Exponential and Rational Quadratic kernel.

---

[3] $\widetilde{O}(\cdot)$ suppresses factors of the form $\log^{O(1)}(n/\varepsilon)$.

An application of our non-negative matrix-vector product result of Theorem 2.6 given in Appendix B.2 is a faster algorithm to (approximately) multiply a kernel matrix with another entry-wise non-negative matrix, for example another kernel matrix.

**Theorem A.2.** *Suppose that for the kernel matrix $K \in \mathbb{R}^{n \times n}$, we have a subroutine which answers a $\varepsilon$-non-negative MVP (Definition 2.1) in time $T(n, \varepsilon)$. Let $A$ be any entry wise non-negative matrix (e.g. another kernel matrix). In time $\widetilde{O}(n \cdot T(n, \varepsilon))$, we can compute a matrix $B$ such that*

$$\|KA - B\|_F \leq \varepsilon \|KA\|_F \leq \varepsilon \cdot \min(\|K\|_2 \cdot \|A\|_F, \|K\|_F \cdot \|A\|_2)$$

*with high probability.*

Note that standard techniques for approximate matrix multiplication multiplication, based on sketching or row sampling (Mahoney et al., 2011; Woodruff et al., 2014; Cohen et al., 2016), roughly say that by sampling $\widetilde{O}(1/\varepsilon^2)$ columns of $K$ and $\widetilde{O}(1/\varepsilon^2)$ rows of $A$, in time $\widetilde{O}(n^2/\varepsilon^2)$ we can compute a $B$ satisfying

$$\|KA - B\|_F \leq \varepsilon \|K\|_F \cdot \|A\|_F.$$

This result is able to replace a frobenius norm factor with a spectral norm factor, e.g. $\|K\|_F$ can be converted to $\|K\|_2$, which can be up to $\Omega(\sqrt{n})$ times smaller.

Now we focus on our main technical result, which is about computing a *relative error* approximation to the top eigenvalue of a kernel matrix $K$, given access to a non-negative matrix-vector product sub-routine of Definition 2.1.

**Theorem A.3.** *Let $\varepsilon \in (0, 1)$. Suppose there exists a $\varepsilon$-non-negative MVP algorithm for a kernel matrix $K$, running in time $T(n, \varepsilon)$. Given this sub-routine, algorithm 2) returns a scalar $\lambda$ and unit vector $u$ satisfying $\lambda_1(K) \geq u^\top K u \geq (1 - (5/8)\varepsilon)\lambda_1(K)$ and $(1 + \varepsilon/8)\lambda_1(K) \geq \lambda \geq (1 - \varepsilon/2)\lambda_1(K)$. The overall runtime of the algorithm is $\widetilde{O}\left(T(n, \varepsilon/8) \cdot \frac{1}{\varepsilon}\right)$.*

Theorem A.3 works in a black-box manner with respect to an underlying $\varepsilon$-non-negative MVP sub-routine. Thus, if we use our improved subroutine of Theorem A.1 for the Gaussian and Laplace kernel case, we get an overall runtime of $\widetilde{O}\left(\frac{dn^{1+p}}{\varepsilon^{3+p}}\right)$ (where again $p$ is their respective value in Table 3). For the Exponential and Rational Quadratic kernel, we can instead use the slower existing subroutine of (Backurs et al., 2021) and obtain an overall runtime of $\widetilde{O}\left(\frac{dn^{1+p}}{\varepsilon^{4+2p}}\right)$ to compute a $1 + \varepsilon$ relative error approximation to $\lambda_1$, where again $p$ is kernel dependent. In comparison, the running time of (Backurs et al., 2021) for obtaining a relative error approximation is $\widetilde{O}\left(\frac{dn^{1+p}}{\varepsilon^{7+4p}}\right)$, in particular, the $\varepsilon$ dependence is $1/\varepsilon^{7+4p}$. Thus for the Gaussian and Laplacian case, we remove a factor of $\approx 1/\varepsilon^{4+3p}$ and for the Exponential and Rational Quadratic kernel, we remove a factor of $\approx 1/\varepsilon^{3+2p}$, where again we note that the $p$ value is kernel dependent. This result is proved in Appendix C.

For the case of computing the sum of all the entries of the kernel matrix up to a $1 + \varepsilon$ relative error, our improved result is the following.

**Theorem A.4.** *Let $k$ be a kernel function admitting a KDE datastructure of Definition A.1 (any in Table 3). Let $K \in \mathbb{R}^{n \times n}$ be the associated kernel matrix for $n$ points in $d$ dimensions, and $\varepsilon \in (0, 1)$. In time $\widetilde{O}\left(\frac{n^{\frac{1}{2}+\frac{p}{2}}}{\varepsilon^4}\right)$, Algorithm 3 the sum of the entries of $K$ up to a $1 + \varepsilon$ factor with probability $\geq 0.99$.*

Fig. 1 shows that we improve over (Backurs et al., 2021) across all the interesting range of $p \leq 1$. We refer to Appendix D.1 for a technical discussion.

# B. Improved approximate Kernel Matrix Vector Multiplication for Gaussian and Laplacian Kernels

The goal of this section is to prove Theorem A.1. First we establish some helpful lemmas.

**Lemma B.1** (Transformation). *Let $k$ be the Gaussian $(e^{-\|x-y\|_2^2})$, Laplacian $(e^{-\|x-y\|_1})$, or the Rational Quadratic $\left(\frac{1}{(1+\|x-y\|_2^2)^\beta}\right)$ kernels. For every $x, y \in \mathbb{R}^d$ and $c \in (0, 1)$, there exists $y' \in \mathbb{R}^d$ such that*

$$c \cdot k(x, y) = k(x', y')$$

*where $x' = [x; 0]$.*

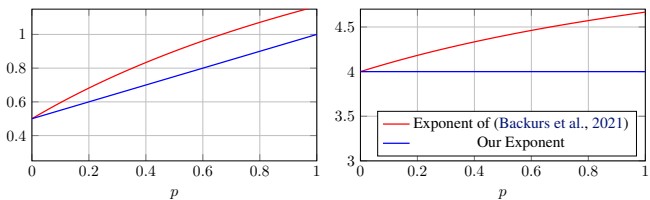

*Figure 1.* Exponent of $n$ (left plot) and exponent of $\varepsilon$ (right plot) for the problem of computing $s(K)$ up to a $1 + \varepsilon$ factor. For the left plot, note that there is never a reason to spend more than $\omega(n)$ time since a simple Chernoff bound calculation shows that sampling $\tilde{O}(n/\varepsilon^2)$ uniformly random entries also suffices to $1 + \varepsilon$ approximate $s(K)$. Thus, we may always take the minimum of the exponent of (Backurs et al., 2021) and 1. Note that the interesting regime of $p$ is $0 \leq p \leq 1$, with the best $p$ values given in Table 3.

**Remark B.1.** Since only the transformation of one of the input points depends on $c$, we say that the weight $c$ is associated with $y$.

*Proof.* For the Gaussian case, write $c = e^{-\log(1/c)}$ and note that $\log(1/c) > 0$. Define $x' = [x; 0], y' = [y; \sqrt{\log(1/c)}]$, which gives

$$c \cdot e^{-\|x-y\|_2^2} = e^{-\left(\|x-y\|_2^2 + \log(1/c)\right)} = e^{-\|x'-y'\|_2^2}.$$

The Laplacian case is similar: $x'$ stays the same but now we let $y' = [y; \log(1/c)]$. $\qquad\square$

The main algorithm in this section is Algorithm 1, which we show admits the following guarantee.

---

**Algorithm 1** Non-negative Matrix Vector Product

---

1: **procedure** NON-NEG-MVP($y \in \mathbb{R}^n, \|y\|_2 = 1, \varepsilon \in (0,1)$)
2:     $b \leftarrow C \log(n/\varepsilon)$ for a sufficiently large constant $C > 1$
3:     Round every coordinate of $y < \frac{\varepsilon}{10n^{3/2}}$ to zero
4:     Partition the coordinates of $y$ into buckets, where the $t$-th bucket $\mathcal{Y}_t$ is for coordinates with value between $2^t/\sqrt{n}$
    (inclusive) and $2^{t+1}/\sqrt{n}$ (exclusive).             ▷ *Note $t$ can be negative.*
5:     $z \leftarrow 0^n$
6:     **for** $i = 1$ to $n$ **do**
7:         **for** $t \in \mathbb{Z}$ **do**                     ▷ *Note we only loop over the non-empty buckets*
8:             $\mu_t \leftarrow \frac{2^{|t|}\varepsilon'}{Cn}$ for a large constant $C > 1$ where $\varepsilon' = \frac{\varepsilon}{b}$
9:             $\mathcal{X}_t \leftarrow \{x_j \in X \mid j \in \mathcal{Y}_t\}$
10:            $C_t \leftarrow$ multiplicative scaling such that all coordinates in $\mathcal{Y}_t$ are in $(0,1)$     ▷ *Note $C_t = \Theta(\sqrt{n}/2^t)$*
11:            Initialize data structure $\mathcal{D}_t$ of Definition A.1 on points $\mathcal{X}_t$, where every $x_j \in \mathcal{X}_t$ is transformed as in
    Lemma B.1 with the weight $C_t y_j$. Set the relative error to $1 + \varepsilon$ and additive error to $\mu_t$.
12:            $z(i) \leftarrow z(i) + |\mathcal{Y}_t| \cdot C_t^{-1} \cdot \mathcal{D}_t([x_i; 0])$
13:         **end for**
14:     **end for**
15:     **Return** $z$ as the approximation to $Ky$
16: **end procedure**

---

**Theorem A.1.** *Let $k$ be the Gaussian ($e^{-\|x-y\|_2^2}$) or the Laplacian ($e^{-\|x-y\|_1}$) kernels and let $p$ be the corresponding exponent in the KDE datastructure stated in Table 3 ($p = 0.173 + o(1)$ and $p = 0.5$ respectively). Let $K \in \mathbb{R}^{n \times n}$ be the associated kernel matrix for $n$ points in $d$ dimensions. There is an $\varepsilon$-non-negative approximate matrix-vector product algorithm (Algorithm 1) satisfying Definition 2.1 for $K$ running in time $\widetilde{O}\left(\frac{dn^{1+p}}{\varepsilon^{2+p}}\right)$.*

The subsequent sections Appendix B.2 and Appendix C analyze applications of this algorithm. See Section 4.1 for an intuitive explanation of the approach. The formal algorithm and proof follow.

In our analysis below, we assume that every query to the KDE data structure satisfies the guarantees of (3). This happens with high probability. The first two lemmas show that removing all sufficiently small coordinates of the input $y$ and then rounding its values has a tolerable effect on the error of the final output.

**Lemma B.2.** *Rounding the coordinates in Step 3 only introduces $\varepsilon \|Ky\|_2$ error.*

*Proof.* Note that

$$\|Ky\|_2^2 \geq \|y\|_2^2 = 1 \tag{4}$$

since the diagonal of $K$ is all ones and the entries of $K - I$ and $y$ are all non-negative. Thus the difference between the original vector and after rounding down just contains all the entries of $y$ that are smaller than $\frac{\varepsilon}{10n^{3/2}}$. Since every entry of $K$ is at most 1, this introduces error at most $\varepsilon/\sqrt{n}$ per coordinate, or an overall error of $\varepsilon \leq \varepsilon \|Ky\|_2$. □

Now let $y^t$ correspond to the entries of $y$ which are in $\mathcal{Y}_t$ (and the rest 0 such that $y = \sum_t y^t$). The following is our main lemma proving our approximation bound.

**Lemma B.3.** *We have $\|Ky - z\|_2 \leq O(\varepsilon) \cdot \|Ky\|_2$.*

*Proof.* Consider a fixed entry $i \in [n]$. Note that the exact value is given by:

$$(Ky^t)(i) = \sum_{\ell \in \mathcal{Y}_t} y_\ell K(x_i, x_\ell) := \Delta$$

In the $i$ case of loop 6 in Algorithm 1, the query $\mathcal{D}_t([x_i; 0])$ satisfies (via Eq. (3))

$$\left| \mathcal{D}_t(x_i) - \frac{C_t y_\ell}{|\mathcal{Y}_t|} \sum_{\ell \in \mathcal{Y}_t} K(x_i, x_\ell) \right| \leq \frac{\varepsilon \cdot C_t y_\ell}{|\mathcal{Y}_t|} \cdot \sum_{\ell \in \mathcal{Y}_t} K(x_i, x_\ell) + \mu_t.$$

Multiplying through by $|\mathcal{Y}_t| \cdot C_t^{-1}$ gives us

$$\left| |\mathcal{Y}_t| \cdot C_t^{-1} \cdot \mathcal{D}_t(x_i) - y_\ell \sum_{\ell \in \mathcal{Y}_t} K(x_i, x_\ell) \right| \leq \varepsilon \Delta + |\mathcal{Y}_t| \cdot C_t^{-1} \cdot \mu_t.$$

We need to bound the additive error in the RHS above. We consider two cases: if $t \geq 0$ (i.e. the coordinates are larger than $1/\sqrt{n}$), then $|\mathcal{Y}_t| \leq n/2^{2t}$ (since the total $\ell_2$ norm of $y$ is 1). This gives us

$$|\mathcal{Y}_t| \cdot C_t^{-1} \cdot \mu_t = O\left( \frac{n}{2^{2t}} \cdot \frac{2^t}{\sqrt{n}} \cdot \frac{2^t \varepsilon'}{n} \right) = O\left( \frac{\varepsilon'}{\sqrt{n}} \right).$$

Otherwise if $t < 0$ (i.e. the coordinates are smaller than $1/\sqrt{n}$), we can only guarantee $|\mathcal{Y}_t| \leq n$, but we know that $C_t^{-1}$ is sufficiently small, giving us

$$|\mathcal{Y}_t| \cdot C_t^{-1} \cdot \mu_t = O\left( n \cdot \frac{1}{2^{|t|}\sqrt{n}} \cdot \frac{2^{|t|} \varepsilon'}{n} \right) = O\left( \frac{\varepsilon'}{\sqrt{n}} \right).$$

Summing across all $t$ (of which there are only $O(b)$ many), we have for every $i \in [n]$, the $i$th entry of $Ky - z$ is bounded in absolute value by $O(\varepsilon) \cdot [Ky](i) + O\left( \frac{\varepsilon}{\sqrt{n}} \right)$, which proves the approximation guarantee.

Finally, we note that we can always easily guarantee the error $e$ is entry-wise non-negative as well since the KDE query of Definition A.1 can be made to always overestimate the value and we can add an extra $O(\varepsilon/\sqrt{n})$ to every coordinate of $z$ to take care of the rounding step we performed in line 3. □

The next lemma bounds the overall running time, including the query and datastructure construction times.

**Lemma B.4.** *Assuming a fast KDE data structure of Definition A.1, the total runtime of Algorithm 1 is bounded by $\widetilde{O}\left( \frac{dn^{1+p}}{\varepsilon^{2+2p}} \right)$.*

*Proof.* We bound the data structure construction and query times separately. First note that for the case of $p > 0$, the sum of $1/\mu^p$ overall $\mu$'s possibly considered in line 8 of Algorithm 1 can be bounded up to constant factors by

$$T = O\left(\sum_{t \geq 0} \left(\frac{n}{2^t \varepsilon'}\right)^p\right) = O\left(\frac{n^p}{\varepsilon'^p} \sum_{t \geq 0} \frac{1}{2^{tp}}\right) = O\left(\frac{n^p}{\varepsilon'^p}\right).$$

Now since $|\mathcal{Y}_i| \leq n$, the total data structure construction times across all instances of $\mathcal{D}_i$ is bounded by

$$\frac{1}{\varepsilon^2} \cdot O\left(\sum_{i=1}^{b} \frac{d|\mathcal{Y}_i|}{\mu_i^p}\right) \leq O\left(\frac{ndT}{\varepsilon^2}\right) = O\left(\frac{n^{1+p}d}{\varepsilon^2 \cdot \varepsilon'^p}\right).$$

Now we bound the total query time. There is always an overhead of $n$ coming from the loop in line 6 of Algorithm 1 since we query each $\mathcal{D}_t$ $n$ times. The cost of a single query summed across all $\mathcal{D}_t$ is bounded by

$$\widetilde{O}\left(\sum_t \frac{d}{\varepsilon^2 \mu_t^p}\right) = \widetilde{O}\left(\frac{dn^p}{\varepsilon^{2+p}}\right),$$

giving a total running time of $\widetilde{O}\left(\frac{dn^{1+p}}{\varepsilon^{2+p}}\right)$, as desired. $\qquad\square$

Putting everything together proves our main theorem.

*Proof of Theorem A.1.* Lemma B.3 shows the desired $\varepsilon \cdot \|Ky\|_2$ error bound (after scaling $\varepsilon$ by a constant) and Lemma B.4 proves the running time, which proves Theorem A.1. $\qquad\square$

**Remark B.2.** Note that the output of Algorithm 1 satisfies a slightly stronger guarantee than stated in Theorem A.1: Every entry of $z$, which is the approximation to $Ky$ for an input non-negative unit vector $y$, satisfies $(Ky)(i) \leq z(i) \leq (1+\varepsilon)(Ky)(i) + O\left(\frac{\varepsilon}{\sqrt{n}}\right)$.

### B.1. Limits of KDE-Based Algorithms for Non-Negative Matrix Vector Product?

We give some evidence that a running time of the form $n^{1+p}$ maybe necessary, if want the 'per coordinate' guarantee of our non-negative matrix vector product gives, namely that every coordinate of $Ky$, for a unit vector $y$, is approximated up to $1 + \varepsilon$ multiplicative error and $\varepsilon/\sqrt{n}$ additive error (see Remark B.2).

**Lemma B.5.** *Suppose there is an algorithm which computes a non-negative matrix vector product on a dataset of size $n$ with the same guarantees as Remark B.2 ($1 + \varepsilon$ multiplicative error and $\varepsilon/\sqrt{n}$ additive error per coordinate) in $T(n, \varepsilon)$ time. Then there exists an algorithm which given a set $X, |X| = n$ and a set of $n$ queries $Z = \{z_1, \cdots, z_n\}$, answers all $n$ KDE queries $\frac{1}{n} \sum_{x \in X} k(z_i, x)$ with $1 + \varepsilon$ multiplicative and $\varepsilon/n$ additive error in $T(O(n), \varepsilon)$ time. That is, the amortized time per KDE query to get a $1 + \varepsilon$ multiplicative and $\varepsilon/n$ additive error is $T(O(n), \varepsilon)/n$ time.*

Before the proof, let's see how the lemma relates to our bound of $\widetilde{O}\left(\frac{dn^{1+p}}{\varepsilon^{2+p}}\right)$ (say for the Gaussian Kernel case). Ignoring logarithmic factors and the $d$ term, our bound can be decomposed into

$$n \times \frac{n^p}{\varepsilon^{2+p}}.$$

Thus the lemma implies that, based on the second term, we can get an amortized KDE query time of $n^p/\varepsilon^{2+p}$ for multiplicative error $1 + \varepsilon$ and relative error $\varepsilon/n$. This is essentially the best KDE query time one can obtain for this desired level of precision (almost by definition from Definition A.1).

Thus one cannot improve upon our algorithm for computing a non-negative matrix vector query, with the same coordinate wise approximation guarantees, unless there is a different family of KDE algorithms different than the form of Definition A.1 (which we are not aware of), or amortization helps (which we are also not aware of). There is one more caveat we must mention: it may also possible to obtain an $\varepsilon$-non-negative matrix vector product without having the stronger coordinate wise guarantee. We leave these as interesting questions for future work.

*Proof of Lemma B.5.* Consider the kernel matrix on the dataset $Z \cup X$ and the query vector $y = [0, \ldots, 0, 1/\sqrt{n}, \ldots, 1/\sqrt{n}]$ where there are $n$ copies of $1/\sqrt{n}$ and $n$ copies of $0$. Then if we compute a non-negative matrix vector product using this query $v$, then the $i$th coordinate of the output is exactly a $1 + \varepsilon$ multiplicative and $\varepsilon/n$ additive approximations to

$$\frac{1}{\sqrt{n}} \sum_{x \in X} k(y_i, x).$$

We need to divide by $1/\sqrt{n}$ to get the correct scaling for a KDE query, so we are left with a $1 + \varepsilon$ multiplicative and $\varepsilon/n$ additive error approximation to each of the $n$ KDE queries of interest, as desired. □

### B.2. Approximate Matrix Multiplication

**Theorem A.2.** *Suppose that for the kernel matrix $K \in \mathbb{R}^{n \times n}$, we have a subroutine which answers a $\varepsilon$-non-negative MVP (Definition 2.1) in time $T(n, \varepsilon)$. Let $A$ be any entry wise non-negative matrix (e.g. another kernel matrix). In time $\widetilde{O}(n \cdot T(n, \varepsilon))$, we can compute a matrix $B$ such that*

$$\|KA - B\|_F \le \varepsilon \|KA\|_F \le \varepsilon \cdot \min(\|K\|_2 \cdot \|A\|_F, \|K\|_F \cdot \|A\|_2)$$

*with high probability.*

*Proof.* We simply view the columns of the matrix $A$ as separate vectors and apply our non-negative matrix vector product guarantee. We have that the $i$th column of our output $B$ is the same as the $i$th column of $KA$, denoted as $(KA)_i$, up to error which has norm at most $\varepsilon \|(KA)_i\|_2$. Then

$$\|B\|_F^2 \le \varepsilon^2 \|(KA)_i\|_2^2 = \varepsilon^2 \|KA\|_F^2,$$

as desired. The last step follows from a well known fact, which we quickly sketch here. Letting $A_i$ denote the $i$th column of $A$,

$$\|K(A_i)\|_2 \le \|K\|_2 \|A_i\|_2 \implies \sum_i \|K(A_i)\|_2^2 \le \|K\|_2^2 \sum_i \|A_i\|_2^2 = \|K\|_2^2 \|A\|_F^2,$$

as desired. The running time follows from $n$ invocation of Theorem A.1. □

## C. Faster Sub-quadratic Time Top Eigenvalue

We now analyze the noisy power method for approximating the top eigenvalue of a given kernel matrix. See Section 4.2 for an intuitive overview of our method.

---

**Algorithm 2** Top Eigenpair Computation

---

1: **Input:** A kernel matrix $K$, a matrix-vector product method `Non-neg-MVP` satisfying Definition 2.1, an error parameter $\varepsilon \in (0, 1)$.
2: **Output:** A scalar $\lambda$ and non-negative unit vector $u$.
3: **procedure** TOP-EIGENPAIR
4:     $T \leftarrow \lceil 10 \log(n)/\varepsilon \rceil$                                          ▷ Number of iterations of power method
5:     Initialize $u = z_0 \leftarrow \frac{1}{\sqrt{n}} \mathbf{1} \in \mathbb{R}^n$
6:     $\lambda \leftarrow z_0^\top K z_0$
7:     **for** $t = 0$ to $T$ **do**
8:         $\widetilde{z}_{t+1} \leftarrow \texttt{Non-neg-MVP}(K, z_t, \varepsilon/8)$                              ▷ Algorithm 1 with precision $\varepsilon/8$
9:         **if** $\langle z_t, \widetilde{z}_{t+1} \rangle > \lambda$ **then**
10:             $u \leftarrow z_t$
11:             $\lambda \leftarrow \langle z_t, \widetilde{z}_{t+1} \rangle$
12:         **end if**
13:         $z_{t+1} \leftarrow \widetilde{z}_{t+1} / \|\widetilde{z}_{t+1}\|$
14:     **end for**
15:     **Return** $\lambda, u$
16: **end procedure**

---

The key lemma is that one of the $z_t$ produced by Algorithm 2 will satisfy a relative error guarantee.

**Lemma C.1.** *Let $z_t$ be defined by $Definition$ 4.1 for $\delta = \varepsilon/8 \leq 0.01$. Let $T = 10\log(n)/\varepsilon$. Then there exists $t \in [0, T]$ for which $z_t^\top K z_t \geq (1 - \varepsilon/2)\lambda_1(K)$.*

*Proof.* By Definition 4.1, we may express the component in the direction of the top eigenvector as

$$\langle v_1, z_{t+1} \rangle = \frac{\langle v_1, \widetilde{z}_{t+1} \rangle}{\|\widetilde{z}_{t+1}\|} = \frac{\langle v_1, K z_t + \|K z_t\| \, \delta u(t) \rangle}{\|\widetilde{z}_{t+1}\|}.$$

Now use that $v_1$ is a left-eigenvector of $K$ and that $\langle v_1, u(t) \rangle \geq 0$ to obtain

$$\langle v_1, z_{t+1} \rangle = \frac{\lambda_1 \langle v_1, z_t \rangle + \delta \|K z_t\| \langle v_1, u(t) \rangle}{\|\widetilde{z}_{t+1}\|} \geq \frac{\lambda_1 \langle v_1, z_t \rangle}{\|\widetilde{z}_{t+1}\|}. \tag{5}$$

The denominator can be upper bounded by the triangle inequality as

$$\|\widetilde{z}_{t+1}\| \leq (1 + \delta) \|K z_t\| \leq (1 + \delta)\sqrt{z_t^\top K z_t \cdot \lambda_1}, \tag{6}$$

where the last inequality follows from writing $z_t$ in the eigenvector basis of $K$ and factoring out $\lambda_1$. Now suppose that

$$z_t^\top K z_t \leq (1 - \varepsilon/2)\lambda_1$$

for all $t \leq T$. Combining this with (5) and (6), we arrive at

$$\langle v_1, z_{t+1} \rangle \geq \frac{\langle v_1, z_t \rangle}{(1 + \delta)\sqrt{1 - \varepsilon/2}}.$$

Since $\langle v_1, z_0 \rangle \geq n^{-1/2}$, we have

$$\langle v_1, z_T \rangle \geq \frac{\langle v_1, z_0 \rangle}{\left((1 + \delta)\sqrt{1 - \varepsilon/2}\right)^T} \geq \frac{n^{-1/2}}{\left((1 + \delta)\sqrt{1 - \varepsilon/2}\right)^T}.$$

Since $\delta = \varepsilon/8 \leq 0.01$, when $T = 10\log(n)/\varepsilon$, we have $\langle v_1, z_T \rangle > 1$ which is a contradiction. $\qquad\square$

The previous lemma shows that there exist a "good" vector $z_t$. The remainder of the proof of our main theorem is that the algorithm will pick up on this property and return it or a similarly good vector.

**Theorem A.3.** *Let $\varepsilon \in (0, 1)$. Suppose there exists a $\varepsilon$-non-negative MVP algorithm for a kernel matrix $K$, running in time $T(n, \varepsilon)$. Given this sub-routine, algorithm 2) returns a scalar $\lambda$ and unit vector $u$ satisfying $\lambda_1(K) \geq u^\top K u \geq (1 - (5/8)\varepsilon)\lambda_1(K)$ and $(1 + \varepsilon/8)\lambda_1(K) \geq \lambda \geq (1 - \varepsilon/2)\lambda_1(K)$. The overall runtime of the algorithm is $\widetilde{O}\big(T(n, \varepsilon/8) \cdot \frac{1}{\varepsilon}\big)$.*

*Proof.* Note that by the guarantee of Definition 2.1, the sequence $z_t$ produced by Algorithm 2 satisfies Definition 4.1. Algorithm 2 returns $\lambda$ and $z_r$ where

$$r = \arg\max_{t \in [0, T]} \langle z_t, \widetilde{z}_{t+1} \rangle, \quad \lambda = \langle z_r, \widetilde{z}_{r+1} \rangle.$$

Note that by Definition 4.1,

$$z_t^\top K z_t \leq \langle z_t, \widetilde{z}_{t+1} \rangle = z_t^\top K z_t + \delta \|K z_t\| \langle z_t, u_t \rangle \leq z_t^\top K z_t + \delta\lambda_1.$$

Let $\alpha = \max_{t \in [0, T]} z_t^\top K z_t$ so that

$$\alpha \leq \max_{t \in [0, T]} \langle z_t, \widetilde{z}_{t+1} \rangle = \lambda \leq \alpha + \delta\lambda_1$$

and

$$\alpha - \delta\lambda_1 \leq z_r^\top K z_r.$$

But Lemma C.1 ensures that
$$(1 - \varepsilon/2)\lambda_1 \le \alpha \le \lambda_1$$

which gives the desired approximation guarantee for $\delta = \varepsilon/8$. The running time of our algorithm can be decomposed into the following:

$$\underbrace{O\left(\frac{\log(n/\varepsilon)}{\varepsilon}\right)}_{\text{\# of iterations}} \times \underbrace{T(n, \varepsilon/8)}_{\text{Running time for a single } \varepsilon/8\text{-precision MVP}} \quad .$$

$\square$

## C.1. Lower Bounds Related to Theorem A.3

In this section, we argue that Theorem A.3 is optimal in a couple different senses. The first is that the precision requirement $\delta$ for each matrix-vector product cannot be much larger than is set by Algorithm 2. The second is that *any* algorithm produces an output based on the $z_t$ requires $\Omega(\log(n)/\varepsilon)$ iterations. Notably, this is *not* the case when $\delta = 0$. When $\delta = 0$, the span of the $z_t$ forms a Krylov space, and so by computing the norm of the compression of $A$ onto this space, one recovers the norm of $A$ in just $\widetilde{O}(1/\sqrt{\varepsilon})$ iterations.

**Proposition C.2** (Optimality of precision). *If $\delta > \varepsilon/(1 - \varepsilon)$, then $\delta$-approximate power iteration does not always result in a unit vector $x$ with $x^\top K x \ge (1 - \varepsilon)\lambda_1(K)$, even for infinitely many iterations.*

*Proof.* We give an example of a $K$ for which power iteration with adversarial noise immediately stagnates, that is, the adversary may cause $z_t = z_0$ for all $t$. Set
$$\lambda = 1 - \frac{n(\varepsilon + \mathrm{d}\varepsilon)}{n - 1}$$

where $\mathrm{d}\varepsilon$ is infinitesimal and consider the matrix $n \times n$

$$K = \begin{bmatrix} 1 \end{bmatrix} \oplus \lambda I.$$

Note that the starting vector $z_0 = \frac{1}{\sqrt{n}}\mathbf{1}$ does not produce the desired approximation as $z_0^\top K z_0 = 1 - (\varepsilon + \mathrm{d}\varepsilon) < 1 - \varepsilon$. Furthermore,
$$z_0 = K z_0 + e$$

when $e = \frac{1-\lambda}{\sqrt{n}}\begin{bmatrix} 0 & 1 & \cdots & 1 \end{bmatrix}^\top$. If $\|e\| \le \delta\|K z_0\|$, then $z_1 = z_0$ is a possible choice for the adversary. To this end, we compute

$$\|e\| = (1 - \lambda) \cdot \sqrt{\frac{n-1}{n}} = (\varepsilon + \mathrm{d}\varepsilon)(1 + O(1/n))$$

and

$$\|K z_0\| = \sqrt{\frac{1}{n} + \frac{n-1}{n}\lambda^2} = (1 - \varepsilon)(1 + O(1/n)).$$

Thus, for $n$ sufficiently large, the ratio $\|e\| / \|K z_0\|$ drops below every value larger than $\varepsilon/(1 - \varepsilon)$, including $\delta$. $\square$

The following proposition shows that any method which uses a Krylov subspace type algorithm, starting with the initial vector of all ones, must use approximately the same number of iterations that our analysis of power method does, showing that the iteration count is nearly tight. Note that if we the matrix vector product was *exact* then we would only need *sublinear* in $1/\varepsilon$ number of iterations (Musco & Musco, 2015), showing a separation between exact and approximate matrix vector product settings.

**Proposition C.3** (Optimality of iteration count). *Any algorithm which estimates the largest eigenvalue of $K$ using the sequence*
$$z_{t+1} = \mathtt{Non\text{-}neg\text{-}MVP}(K, z_t, \delta)$$

*for $z_0 \in \mathrm{span}(\mathbf{1})$ must use*
$$\frac{\log(\frac{\delta}{2\varepsilon}\sqrt{n})}{\log(1 + 2\varepsilon)} = \Omega\left(\frac{\log((\delta/\varepsilon)^2 n)}{\varepsilon}\right)$$

*iterations.*

*Proof.* Let $K = I$ and $K' = I + 2\varepsilon e_1 e_1^\top$. Let $z_t, z_t'$ be defined by $z_0 = z_0' = \mathbf{1}$ and

$$z_{t+1} = \mathtt{NN\text{-}MVP}(K, z_t) \quad \& \quad z_{t+1}' = \mathtt{NN\text{-}MVP}(K', z_t').$$

Then the adversary may make

$$z_t = z_t' \quad \forall t \leq \log(\sqrt{n}/2)/\log(1 + 2\varepsilon).$$

We prove this by induction. Note the adversary can make $z_{t+1}' = K' z_t'$ exactly. Suppose $z_t = z_t'$. Then

$$\begin{aligned} z_{t+1} &= K z_t' + e \\ &= (K' - 2\varepsilon e_1 e_1^\top) z_t' + e \\ &= z_{t+1}' - 2\varepsilon e_1 e_1^\top z_t' + e. \end{aligned} \tag{7}$$

Now the adversary may set $e = 2\varepsilon e_1 e_1^\top z_t' = 2\varepsilon(1 + 2\varepsilon)^t e_1$ whenever $2\varepsilon(1 + 2\varepsilon)^t \leq \delta\sqrt{n}$, which is equivalent to $t \leq \log(\frac{\delta}{2\varepsilon}\sqrt{n})/\log(1 + 2\varepsilon)$. Thus, the algorithm must give the same output on input $K$ and $K'$, but the range of acceptable outputs for those inputs are disjoint. $\qquad\square$

**The non-negativity property.** In our analysis, we crucially used the fact that the top eigenvector $v_1$ has non-negative entries and that the error vector $e$ can be made to have non-negative entries. This implies that all times $t$, $e(t)^\top v_1 \geq 0$. Intuitively, this means that the adversary can only 'cancel out' the contribution of the top eigenvalue by boosting coordinates associated with other eigenvalues; it cannot directly reduce the coordinate associated with the top eigenvalue. This fact is crucially used throughout our analysis. Here we give a simple example showing why without this property, power method cannot work in our (adversarial) noise setting, unless the precision $\delta$ is extremely small.

**Example 2.**

$$K = \begin{pmatrix} 1 & & & \\ & 1/2 & & \\ & & \ddots & \\ & & & 1/2 \end{pmatrix} \in \mathbb{R}^{n \times n}$$

and let the starting vector in the power method be $\left( \frac{1}{\sqrt{n}} \quad \cdots \quad \frac{1}{\sqrt{n}} \right)$. Then

$$Kx = \left( \frac{1}{\sqrt{n}} \quad \frac{1}{2\sqrt{n}} \quad \cdots \quad \frac{1}{2\sqrt{n}} \right).$$

Now suppose we no longer have the property that $e(t)^\top v_1 \geq 0$. Then if $\delta = \omega(1/\sqrt{n})$, then in the *very next* iteration, the noisy power method can instead return the vector

$$\left( 0 \quad \frac{1}{2\sqrt{n}} \quad \cdots \quad \frac{1}{2\sqrt{n}} \right),$$

which satisfies the requirement that the norm of the error is bounded by $\delta\|Kx\|_2 = O(\delta)$. However, now the resulting vector lies completely in the orthogonal complement of the top eigenvector, so no additional iterations of the power method, even without any noise, can give a $1 - \varepsilon$ approximation of the top eigenvalue.

## D. Improved Vector-Matrix-Vector Product and Kernel Sum

An immediate application of our faster non-negative matrix-vector product is a fast algorithm for estimating a vector-matrix-vector product $v^\top K v$ for an entry-wise non-negative vector $v$. Note that we must read all the entries of the vector $v$ (otherwise we cannot detect the all zeros vector versus say a basis vector), meaning that the problem has a $\Omega(n)$ lower bound.

**Theorem D.1.** *Let $\varepsilon \in (0, 1)$. Suppose there exists a $\varepsilon$-non-negative MVP algorithm for a kernel matrix $K$, running in time $T(n, \varepsilon)$. Let $v \in \mathbb{R}^n$ be an entry-wise non-negative vector. We can compute $v^T K v$ up to a $1 + \varepsilon$ multiplicative factor in time $O(T(n, \varepsilon) + n)$.*

*Proof.* Without loss of generality, assume $v$ is a unit vector. First note that

$$v^\top K v = v^\top (K - I)v + \|v\|_2^2 \geq 1,$$

since all the entries of $K - I$ are non-negative. We simply compute an approximation $y$ to the matrix vector product $Kv$ using the algorithm of Theorem A.1, where $y = Kv + e$ where $e$ is entry-wise non-negative and $\|e\|_2 \leq \varepsilon\|Kv\|_2$. Then we compute $v^\top Ky$. The running time is dominated by the matrix-vector product, which requires $\widetilde{O}\left(\frac{dn^{1+p}}{\varepsilon^{2+2p}}\right)$ time from Theorem A.1.

For the approximation guarantee, note that $v^\top Ky = v^\top Kv + v^\top e$, where $v^\top e \geq 0$. We additionally have (see Remark B.2),

$$v^\top e = \sum_{i=1}^n v(i)e(i) \leq \sum_{i=1}^n v(i)\left(\varepsilon v(i) + O\left(\frac{\varepsilon}{\sqrt{n}}\right)\right) \leq \varepsilon + O\left(\frac{\varepsilon}{\sqrt{n}}\right)\sum_{i=1}^n v(i) \leq O(\varepsilon) \leq O(\varepsilon) \cdot v^\top Kv,$$

implying that $v^\top Ky \in (1 \pm O(\varepsilon))v^\top Kv$, as desired.

$\square$

## D.1. Improved Kernel Matrix Sum

In this section, we give a more involved algorithm for computing $\mathbf{1}^\top K \mathbf{1}$, which is the sum of the entries of the kernel matrix. Recall our notation that $s(K) = \mathbf{1}^\top K \mathbf{1}$ and $s_o(K)$ denotes the off-diagonal sum. Note that the almost trivial $\Omega(n)$ lower bound of computing a general $v^\top Kv$ does not apply here since we already know what all the entries of $\mathbf{1}$ are. Indeed, in this special but practically relevant case (e.g. it can be used to compute the *kernel alignment* problem, a similarity measure between kernel matrices themselves in practice (Cristianini et al., 2001), and it is a commonly used U-statistic (Anastasiou et al., 2023)), much faster algorithms are possible, as discussed in Section 2. Our main theorem is the following:

**Theorem A.4.** *Let $k$ be a kernel function admitting a KDE datastructure of Definition A.1 (any in Table 3). Let $K \in \mathbb{R}^{n \times n}$ be the associated kernel matrix for $n$ points in $d$ dimensions, and $\varepsilon \in (0, 1)$. In time $\widetilde{O}\left(\frac{n^{\frac{1}{2}+\frac{p}{2}}}{\varepsilon^4}\right)$, Algorithm 3 the sum of the entries of $K$ up to a $1 + \varepsilon$ factor with probability $\geq 0.99$.*

See Section 4.3 for an intuitive overview of the proof.

### D.1.1. MAIN SAMPLING LEMMA

In this subsection, we prove our main sampling workhorse lemma. For a subset $A \subseteq [n]$, let $K_A$ be the principal submatrix of the kernel matrix where we keep the rows and columns indexed by $A$. The following lemma generalizes Lemma 6 in (Backurs et al., 2021).

**Lemma D.2.** *Let $K$ be a symmetric (not necessarily kernel) matrix with all entries in $[0, \alpha]$ for a parameter $0 \leq \alpha \leq 1$. Further suppose that all the row and column sums of $K$, ignoring the diagonal, are bounded by a parameter $\beta \geq 0$. Let $A \subseteq [n]$ be a random set where we pick every element of $[n]$ to be in $A$ with probability $q$ independently. Let $Z = \operatorname{diag}(K) + s_o(K_A)/q^2$, where $s_o(K_A)$ denotes the sum of the off diagonal entries of $K_A$, and $\operatorname{diag}(K)$ denotes the sum of the diagonal of $K$. Then*

- $\mathbb{E}[Z] = s(K)$,

- $\operatorname{Var}[Z] \leq O(\alpha q^{-2} + \beta q^{-1}) \cdot s(K)$.

*Proof.* The expectation is clear so we just have to bound the variance. Since variance is preserved under additive shifts, we instead consider $\operatorname{Var}[Z - \operatorname{diag}(K)] = \operatorname{Var}[s_o(K_A)/q^2]$. The expectation of the square is:

$$\mathbb{E}\left[\left(\frac{s_o(K_A)}{q^2}\right)^2\right] = \frac{1}{q^4}\left(q^2\sum_{i \neq j} K_{i,j}^2 + 2q^3 \sum_{|\{i,j,j'\}|=3} K_{i,j}K_{i,j'} + q^4 \sum_{|\{i,j,i',j'\}|=4} K_{i,j}K_{i',j'}\right). \tag{8}$$

We bound each term separately. The first term satisfies

$$q^{-2}\sum_{i \neq j} K_{i,j}^2 \leq q^{-2}\alpha\sum_{i,j} K_{i,j} = q^{-2}\alpha s(K).$$

For the third term, we have

$$\sum_{|\{i,j,i',j'\}|=4} K_{i,j} K_{i',j'} \leq \sum_{i,j,i',j', i\neq j, i'\neq j'} K_{i,j} K_{i',j'} \leq (s(K) - \text{diag}(K))^2.$$

Since

$$\mathbb{E}[s_o(K_A)/q^2]^2 = (s(K) - \text{diag}(K))^2,$$

this term directly cancels with the third term in Eq. (8) in the variance calculation. Thus, it remains to bound the second term in Eq. (8) We have

$$\sum_{|\{i,j,j'\}|=3} K_{i,j} K_{i,j'} \leq O\left(\sum_i s_{o,i}(K)^2\right) \leq O(\max_i s_{o,i}(K) \cdot s(K)),$$

where $s_{o,i}(K)$ denotes the sum of the $i$th row of $K$, ignoring the diagonal. This is because every entry in $K$ is only multiplied by other entries sharing the same row and column, and $K$ is symmetric. This implies that the second term in Eq. (8) can be bounded, up to constant factors, by

$$q^{-1} \cdot \max_i s_{o,i}(K) \cdot s(K) \leq q^{-1} \beta s(K),$$

by our definition of $\beta$. The lemma follows from putting together our calculations. $\qquad \square$

A useful corollary is the case of the previous lemma applied to a kernel matrix $K$.

**Corollary D.3.** *Consider the setting of Lemma D.2 and let $K$ be be a kernel matrix (as defined in Section 3) with $q = C(\varepsilon^2 \sqrt{n})^{-1}$ for a large constant $C \geq 1$. Consider the random variable $Z = n + s_o(K_A)/q^2$ from Lemma D.2. We have*

- $\mathbb{E}[Z] = s(K)$,

- $\text{Var}[Z] \leq 0.001 \cdot \varepsilon^2 \cdot s(K)^2$.

*Proof.* We set $\alpha = 1$ in Lemma D.2. Now Lemma 5 in (Backurs et al., 2021) gives

$$\max_i s_{o,i}(K) \leq O(\sqrt{n} + \sqrt{s_o(K)}),$$

where $s_o(K) = \sum_i s_{o,i}(K)$ denotes the sum of all the off diagonal entries of $K$. Denote this quantity by $\beta$. We have

$$\alpha q^{-2} + \beta q^{-1} \leq O(\varepsilon^4 n + \varepsilon^2 \sqrt{n} \cdot (\sqrt{n} + \sqrt{s_o(K)})) = O(\varepsilon^4 n + \varepsilon^2 n + \varepsilon^2 \sqrt{ns(K)}).$$

Using the fact $s(K) \geq n$ gives us that the above quantity is bounded by $O(\varepsilon^2 s(K))$, so $(\alpha q^{-2} + \beta q^{-1}) \cdot s(K) \leq O(\varepsilon^2 s(K)^2)$. Increasing the constant $C$ in the definition of $q$ as necessary completes the proof. $\qquad \square$

### D.2. A Faster Algorithm for the Kernel Sum

Now we present our faster algorithm for computing $s(K)$, the sum of all the entries of the kernel matrix, up to $1 + \varepsilon$ multiplicative factor.

---

**Algorithm 3** Faster algorithm for Kernel Sum

---

1: **procedure** FAST-KERNELSUM($X = \{x_1, \ldots, x_n\} \subset \mathbb{R}^d$)

2:     $q_1 \leftarrow \min\left(\frac{C}{\varepsilon^2 \sqrt{n}}, 1\right)$ for a large constant $C \geq 1$

3:     Sample every $i \in [n]$ with probability $q_1$ to form subset $A = \{a_1, \ldots, a_m\} \subseteq \{x_1, \ldots, x_n\}$. ▷ *Defines a principal submatrix $K_A$ of $K$.*

4:     Let $B = \{a_i \in A \mid \mathcal{D}_{A \setminus a_i}(a_i) \geq \tau\}$, where we set $\tau = C/(m\varepsilon^3)$ and additive error $\mu = \varepsilon\tau/C$ in the construction of the KDE datastructure ▷ *Note the query $\mathcal{D}_{A \setminus a_i}(a_i)$ **excludes** $a_i$. Remark D.1 details how this can be achieved with a logarithmic overhead via a standard trick.*

5:     $\hat{S}_1 \leftarrow \sum_{b \in B} \mathcal{D}_{B \setminus b}(b)$ where we set additive error $\mu$                     ▷ `Estimate of` $s_o(K_B)$

6:     $\hat{S}_2 \leftarrow \sum_{b \in B} \mathcal{D}_{A \setminus b}(b)$ where we set additive error $\mu$               ▷ `Estimate of` $s_o(K_{B \times A})$

7:     $\hat{S}_3 \leftarrow 2\hat{S}_2 - \hat{S}_1$

8:     $q_2 \leftarrow \min(C\varepsilon^{1.5}\sqrt{m\tau}, 1)$ for $\tau$ defined line Line 4

9:     Sample every row and column in $K_{A \setminus B}$ with probability $q_2$ to form subset $B' \subseteq A \setminus B$

10:     $\hat{S}_4 \leftarrow \sum_{b' \in B'} \mathcal{D}_{B' \setminus b'}(b')$ with KDE additive error $\mu' = \widetilde{O}\left(\frac{\sqrt{\tau}}{\varepsilon^{1.5}\sqrt{m}}\right)$     ▷ `Estimate of` $s_o(K_{B'})$

11:     **Return** $n + q_1^{-2} \cdot (\hat{S}_3 + q_2^{-2} \cdot \hat{S}_4)$ as the approximation to $s(K)$, the sum of entries of $K$

12: **end procedure**

---

*Proof of Theorem A.4.* First we show the approximation guarantee. The proof will be broken down in three steps which we outline before presenting the full details. The first step samples a $\approx \frac{\sqrt{n}}{\varepsilon^2} \times \frac{\sqrt{n}}{\varepsilon^2}$ principal submatrix $K_A$ of $K$, and we show that it suffices to approximate the off-diagonal sum $s_o(K_A)$ up to $1 + \varepsilon$ multiplicative and poly$(1/\varepsilon)$ additive error. To approximate $s_o(K_A)$, we divide the rows and columns of $K_A$ into "heavy" and "light" groups, where heavy means their off-diagonal sum is at least $m\tau$. Thus, the second step removes all the heavy rows and columns of $K_A$, detected using appropriately-tuned KDE queries, and computes a $1 + \varepsilon$ multiplicative approximation of their contribution to $s_o(K_A)$. Finally, we need to deal with the light principal submatrix left. Here, we have the guarantee that their row and column sums are bounded by $m\tau$. So in the third step, we perform another sampling to substantially reduce the size of the light submatrix, which is only possible due to the bounded row/column sum guarantee. Crucially, the sub-sampled matrix remains square, which allows us to take full advantage of KDE queries again.

**Step 1.** Define the random variable $Z = n + s_o(K_A)/q_1^2$, where $K_A$ is the $m \times m$ matrix with rows and columns indexed by the random set $A = \{a_1, \ldots, a_m\}$ sampled in Line 3 of Algorithm 3, and $s_o(\cdot)$ denotes the off-diagonal sum. Corollary D.3, along with Chebyshev's inequality, implies that

$$|Z - \mathbb{E}[Z]| = |Z - s(K)| \leq \varepsilon s(K),$$

with probability $\geq 99\%$. We condition on this event for the rest of the proof. Since $1/q_1^2 = O(\varepsilon^4 n)$, it suffices to estimate $s_o(K_A)$ up to a multiplicative $1 + \varepsilon$ factor and additive error $O(1/\varepsilon^3)$. The latter is because this will imply our additive error in computing $Z$ is bounded by $O\left(\frac{1}{q_1^2} \cdot \frac{1}{\varepsilon^3}\right) \leq O(\varepsilon n) \leq O(\varepsilon s(K))$, noting that $s(K) \geq n$ due to the diagonal.

Thus, for the rest of the proof, we will show that the algorithm estimates $s_o(K_A)$ up to $1 + \varepsilon$ multiplicative factor and additive error $O(1/\varepsilon^3)$ (by increasing the constant $C$ in $q_1$, we can make the additive error any small constant multiple of $1/\varepsilon^3$).

**Step 2.** Towards this end, we have

$$s_o(K_A) = s_o(K_B) + s_o(K_{A \setminus B}) + 2s,$$

where $s_o(K_B)$ is the sum of the off-diagonal entries of the submatrix $K_B$ and similarly $s_o(K_{A \setminus B})$ is the sum of the off-diagonal entries of the submatrix $K_{A \setminus B}$. The $2s$ term comes from the two rectangles left over from carving out $K_B$ and $K_{A \setminus B}$ from $K_A$ (see Figure 2).

Note that $B$ contains all the rows of $K_A$ where the off-diagonal sum is at least $m\tau/2$, for $\tau$ defined in Line 4 of Algorithm 3, since the additive error in the KDE estimate is $O(\varepsilon\tau)$ (and we multiply by $m$ since a KDE query returns an approximation to the scaled sum). Thus, every row of $K_A$ in the set $A \setminus B$ must also have its off-diagonal sum bounded by $m\tau/2$. In particular, if we consider the submatrix $K_{A \setminus B}$, every off-diagonal row and column sum of this matrix is bounded by $m\tau/2$.

Now we show how to obtain a $1 + \varepsilon$ approximation to $s_o(K_B)$ and $2s$. Note that $\sum_{a \in B} \mathcal{D}_{A \setminus a}(a)$, where the additive error in the KDE queries are all set to $\mu$ as defined in Line 4 of Algorithm 3, is a $1 + O(\varepsilon)$ approximation to $s_o(K_B) + s$. This is because the additive error $\mu$ is at most $O(\varepsilon)$ times the sum of the off-diagonal values of the rows in $B$, since $B$ consists of all the $a_i \in A$ where $\mathcal{D}_{A \setminus a_i}(a_i) \geq \tau$, meaning the additive error can be absorbed into the multiplicative one. Furthermore, the sum $\sum_{b \in B} \mathcal{D}_{B \setminus b}(b)$, where again we use the same additive error $\mu$, is a multiplicative $1 + \varepsilon$ and additive $\mu|B|$ approximation to $s_o(K_B)$, and thus

$$2 \cdot \left( \sum_{a \in B} \mathcal{D}_{A \setminus a}(a) \right) - \sum_{b \in B} \mathcal{D}_{B \setminus b}(b)$$

is a $1 + O(\varepsilon)$ multiplicative approximation to $s_o(K_B) + 2s$, since the additive error $O(\mu|B|m) = O(\varepsilon\tau|B|m)$ is at most $\varepsilon$ fraction of $\Omega(\tau|B|m) \leq s_o(K_B) + s \leq s_o(K_B) + 2s$.

**Step 3.** It remains to deal with the term $s_o(K_{A \setminus B})$. Towards that end, note that all the rows and columns of $K_{A \setminus B}$ have their entries bounded by $O(m\tau)$ (since we removed all "large" rows and columns in $B$). Then Lemma D.2 implies that if we sample the rows and columns of $K_{A \setminus B}$ with probability $q_2$ (as defined in Line 8 of Algorithm 3) and compute the (scaled) sum of off-diagonal entries of the resulting matrix $K'$, then

$$\mathrm{Var}\left[ \frac{s_o(K')}{q_2^2} \right] \leq O(m\tau q_2^{-2} + m\tau q_2^{-1}) \cdot s_o(K_{A \setminus B}),$$

where we upper bound both $\alpha = \beta = O(m\tau)$ in the lemma statement and note that we can ignore the diagonal contributions since we are effectively setting the diagonal as zeros by only considering the off-diagonal sum. We have

$$O(m\tau q_2^{-2} + m\tau q_2^{-1}) \cdot s_o(K_{A \setminus B}) = O(m\tau q_2^{-2}) \cdot s_o(K_{A \setminus B}) \leq O\left( \frac{s_o(K_{A \setminus B})}{\varepsilon^3} \right).$$

By Chebyshev's inequality, we have

$$\Pr\left( \left| \frac{s_o(K')}{q_2^2} - s_o(K_{A \setminus B}) \right| \geq \varepsilon s_o(K_{A \setminus B}) + \frac{1}{\varepsilon^3} \right) \leq \frac{\mathrm{Var}\left[ \frac{s_o(K')}{q_2^2} \right]}{\left( s_o(K_{A \setminus B}) + \frac{1}{\varepsilon^3} \right)^2}$$

$$\leq \frac{\varepsilon^3 \mathrm{Var}\left[ \frac{s_o(K')}{q_2^2} \right]}{2 s_o(K_{A \setminus B})},$$

where the last step is due to

$$\left( s_o(K_{A \setminus B}) + \frac{1}{\varepsilon^3} \right)^2 \geq \frac{2 s_o(K_{A \setminus B})}{\varepsilon^3}.$$

Plugging in

$$\mathrm{Var}\left[ \frac{s_o(K')}{q_2^2} \right] \leq O\left( \frac{s_o(K_{A \setminus B})}{\varepsilon^3} \right),$$

and adjusting the constant $C$ in the definition of $q_2$ to be sufficiently large, shows that

$$\left| \frac{s_o(K')}{q_2^2} - s_o(K_{A \setminus B}) \right| \leq \varepsilon s_o(K_{A \setminus B}) + \frac{1}{\varepsilon^3}$$

with probability at least $0.999$.

Thus, it suffices to compute the value $s_o(K')$. We will do this with additive error $\mu' = \widetilde{O}\left( \frac{\sqrt{\tau}}{\varepsilon^{1.5}\sqrt{m}} \right)$. With this setting, the overall additive error in computing $s_o(K')$ is $\mu'm'$ and we have

$$\frac{\mu'm'}{q_2^2} \leq \widetilde{O}\left( \frac{\sqrt{\tau}}{\varepsilon^{1.5}\sqrt{m}} \cdot \frac{m'}{\varepsilon^3 m\tau} \right) = \widetilde{O}\left( \frac{1}{\varepsilon^3} \cdot \frac{m'}{\varepsilon^{1.5}m\sqrt{m\tau}} \right),$$

and we exactly have $m' = \widetilde{O}(\varepsilon^{1.5} m \sqrt{m\tau})$ with high probability. Thus by decreasing the value of $\mu'$ by logarithmic factors, we have

$$\frac{\mu' m'}{q_2^2} \leq \frac{1}{\varepsilon^3},$$

so this value of $\mu'$ suffices for an additive $1/\varepsilon^3$ error (which contributes to a $1/\varepsilon^3$ additive error for our estimate of $s_o(K_A)$ which we can tolerate).

**Putting Everything Together.** We know $s_o(K_A) = s_o(K_B) + s_o(K_{A \setminus B}) + 2s$, and all three terms are non-negative. We obtained a $1 + \varepsilon$ multiplicative approximation to $s_o(K_B) + 2s$ in Step 2 and a $1 + \varepsilon$ multiplicative and $1/\varepsilon^3$ additive approximation to $s_o(K_{A \setminus B})$ in Step 3, which gives us a $1 + \varepsilon$ multiplicative and $1/\varepsilon^3$ additive approximation to $s_o(K_A)$, as desired. This proves the approximation guarantee.

**Proving the Running Time.** For the running time, we know Step 2 takes $\widetilde{O}\left(\frac{m}{\varepsilon^2 (\varepsilon\tau)^p}\right)$ time. For Step 3, $K'$ is a $m' \times m'$ matrix with $m' = \widetilde{O}(\varepsilon^{1.5} m \sqrt{m\tau})$ with high probability. We compute the sum of the entries of $K'$ using $m'$ KDE queries with additive error $\mu' = \widetilde{O}\left(\frac{\sqrt{\tau}}{\varepsilon^{1.5}\sqrt{m}}\right)$, giving a runtime of

$$\widetilde{O}\left(\frac{m'}{\varepsilon^2 (\mu')^p}\right) = \widetilde{O}\left(m^{p/2+3/2} \tau^{1/2-p/2} \varepsilon^{3p/2-1/2}\right).$$

To balance, if we set the expressions

$$m^{p/2+3/2} \tau^{1/2-p/2} \varepsilon^{3p/2-1/2} = \frac{m}{\varepsilon^2 (\varepsilon\tau)^p},$$

where the first term corresponds to the running time of computing $s_o(K_{A \setminus B})$ and the second term corresponds to the running time of computing $s_o(K_B)$ (ignoring logarithmic factors), this would solve to

$$\frac{1}{\tau} = \left(m^{1/2+p/2} \varepsilon^{3p/2+3/2}\right)^{\frac{2}{p+1}} = m\varepsilon^3,$$

which is (asymptotically) our choice of $\tau$. Thus, neither term dominates, meaning that the runtime can be bounded by

$$\widetilde{O}\left(\frac{m}{\varepsilon^{2+p}} \cdot \left(m^{1/2+p/2} \varepsilon^{3p/2+3/2}\right)^{\frac{2p}{p+1}}\right) = \widetilde{O}\left(\frac{m^{1+p}}{\varepsilon^{2-2p}}\right).$$

Noting that $m = \Theta(\sqrt{n}/\varepsilon^2)$ with high probability, we have that the total runtime is bounded by

$$\widetilde{O}\left(\frac{n^{\frac{1}{2}+\frac{p}{2}}}{\varepsilon^4}\right).$$

Finally, note that by using the standard trick of repeating $O(\log n)$ times and taking the median, we can boost the success probability of Algorithm 3 to $1 - 1/n^{\Theta(1)}$.

$\square$

**Remark D.1.** We briefly mention how to construct a KDE datastructure $\mathcal{D}'$ over a dataset $X = \{x_1, \ldots, x_n\}$ (with some desired relative error $1 + \varepsilon$ and additive error $\mu$), capable of answering queries $\mathcal{D}_{X \setminus x_i}(y)$ for a given query $y$ and $x_i \in X$. In other words, the sum excludes any desired vector $x_i$. Note that this same construction appears in many prior works, including (Backurs et al., 2021) and (Bakshi et al., 2023), but we briefly mention it here for completeness.

Assume without loss of generality that $n$ is a power of 2. Let $\mu' = \mu/(100 \log n)$. We build a data structure for points $x_1, \ldots, x_{n/2}$ and another for $x_{n/2+1}, \ldots, x_n$, both with additive error $\mu'$. We additionally build 4 data structures for sets $\{x_1, \ldots, x_{n/4}\}, \{x_{n/4+1}, \ldots, x_{n/2}\}, \{x_{n/2+1}, \ldots, x_{3n/4}\}$, and $\{x_{3n/4+1}, \ldots, x_n\}$. We continue this pattern for $O(\log n)$ levels which naturally corresponds to a binary tree with the interval $\{1, \cdots, n\}$ at the top and every interval has two children representing its first and second half. Then to estimate the sum $\frac{1}{n} \sum_{j=1, j \neq i}^{n} k(y, x_j)$ for a given query $y$ and index $i$ to exclude, we represent $\{1, \ldots, i-1\} \cup \{i+1, \ldots, n\}$ as the disjoint union of $O(\log n)$ intervals represented in the binary tree. All these values are positive and the total additive error is $O(\mu' \cdot \log n) \leq \mu$. It can be easily computed that the total construction and query times only blow up by $\text{poly}(\log n)$ factors.

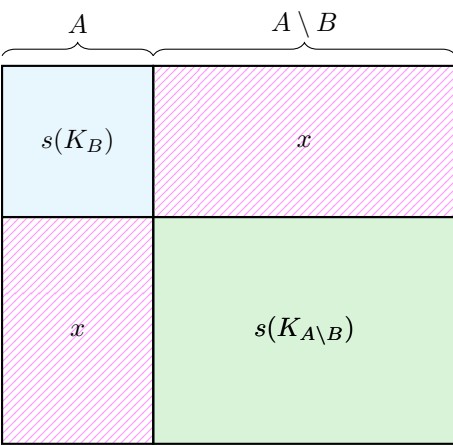

*Figure 2.* In Algorithm 3, we need to compute the sum $s_o(K_A)$, which can be broken down as $s_o(K_B) + s_o(K) + 2x$, where $x$ represents one of the pink rectangles.

### D.3. Limits of KDE-Based Algorithms for the Kernel Sum?

We briefly argue why $\Omega(n^{1/2+p/2}/\varepsilon^{4-p})$ is a natural limit of our ideas. Theorem 2.5 shows that we must sample $m = \Omega(\sqrt{n}/\varepsilon^2)$ points in the input set $X$; otherwise we have no hopes of obtaining a $1 + \varepsilon$ multiplicative approximation. However, this is only a sample complexity lower bound. We now argue how the extra term $n^{p/2}/\varepsilon^{2-p}$ is a natural limitation for processing these $m$ sampled points.

After we sample, obtaining a $1 + \varepsilon$ multiplicative approximation is equivalent to obtaining a $1/\varepsilon^3$ additive error of the off diagonal entries of the sampled set, since we multiply the estimator by $\varepsilon^4 n$ and the original kernel sum is at least $n$. Thus, we must detect every row with off-diagonal sum at least $1/\varepsilon^3$, otherwise our additive error is already off by too much. The only reasonable way to do this seems to be test every row with a KDE query, which requires setting $\mu = 1/(m\varepsilon^3) \leq 1/(\sqrt{n}\varepsilon)$, giving a running time of at least $n^{p/2}\varepsilon^p/\varepsilon^2$ per row, (using our definition of a KDE query in Definition A.1) and an overall running time of at least $n^{1/2+p/2}/\varepsilon^{4-p}$. Thus, to improve upon our result by more than an $\varepsilon^p$ factor, one would need a substantially different algorithm for detecting all rows with sum larger than $1/\varepsilon^3$, besides computing $m$ KDE queries with the appropriate additive error.

### D.4. Optimal Sampling Lower Bound

Our algorithm for computing $\mathbf{1}^\top K \mathbf{1}$ (Algorithm 3) samples $\Theta(\sqrt{n}/\varepsilon^2)$ data points to compute the sum, which we show is optimal. This is a strengthening of a similar result in (Backurs et al., 2021) which proved a $\Omega(\sqrt{n})$ lower bound.

We consider the following distribution $\mathcal{D}(p)$ over vectors in $\mathbb{R}^n$ parameterized by $p \in [0, 1]$:

- With probability $1 - p$, return $n^{100}$ times a uniformly random basis vector.

- With probability $p$, return the origin $0$.

The same proof below applies to the Exponential, Gaussian, and Laplacian kernels.

**Lemma D.4.** *Let $p_1 = (1 + \varepsilon)/\sqrt{n}, p_2 = 1/\sqrt{n}$ and assume $n$ is sufficiently large. Consider the kernel matrices $K_1$ and $K_2$ corresponding to $n$ points drawn from $\mathcal{D}(p_1)$ and $\mathcal{D}(p_2)$ respectively. Let $s(\cdot)$ denote the sum of the entries. With probability at least $1 - \exp(-\Omega(\varepsilon^2 \sqrt{n}))$, we have:*

1. $\mathbb{E}[s(K_1)] \geq (3 + \varepsilon/2)n.$

2. $\mathbb{E}[s(K_2)] \leq (3 + \varepsilon/4)n.$

3. $\Pr(|s(K_i) - \mathbb{E}[s(K_i)]| \geq \varepsilon n/100) \leq 10^{-3}.$

*Proof.* Among the $n$ points (either from $\mathcal{D}(p_1)$ or $\mathcal{D}(p_2)$), for $x \in \{0, e_1, \ldots, e_n\}$, let $c_x$ denote the number of points in the $n$ points that are copies of $x$. Then the kernel sum is equal to

$$\sum_{x \in \{0, e_1, \ldots, e_n\}} c_x^2 + o(e^{-n}).$$

Thus,

$$
\begin{aligned}
\mathbb{E}[s(K_i)] &= n \, \mathbb{E}[c_{e_1}^2] + \mathbb{E}[c_o^2] + o(e^{-n}) \\
&= n^2(n-1)\left(\frac{1-p_i}{n}\right)^2 + n(1-p_i) + n(n-1)p_i^2 + np_i + o(e^{-n}) \\
&= 2n + n^2 p_i^2 + o(n),
\end{aligned}
$$

and the first two claims follow.

We now show concentration of the random variable $\sum_{x \in \{0, e_1, \ldots, e_n\}} c_x^2$. For $x \in \{e_1, \ldots, e_n\}$, write

$$c_x^2 = c_x^2 \cdot \mathbf{1}\{c_x \leq 100 \log n\} + c_x^2 \cdot \mathbf{1}\{c_x > 100 \log n\}.$$

Then since each fixed $c_x$ is a binomial random variable, the Chernoff bound implies that in either case ($i = 0$ or $i = 1$),

$$\Pr(\exists c_x > 100 \log n) \leq n \Pr(c_1 > 100 \log n) \leq n \cdot e^{-50\sqrt{n} \cdot \frac{1}{\sqrt{n}}} < \frac{1}{n^{10}}.$$

Thus,

$$\mathbb{E}\left[\sum_x c_x^2 \cdot \mathbf{1}\{c_x > 100 \log n\}\right] = o(1),$$

and the probability that $\sum_x c_x^2 \cdot \mathbf{1}\{c_x > 100 \log n\}$ deviates by more than $\varepsilon n/200$ from its expected value is at most $1/n$ by Markov's inequality.

To handle $c_x^2 \cdot \mathbf{1}\{c_x \leq 100 \log n\}$, we note that changing any one of the sampled points can only change $\sum_x c_x^2 \cdot \mathbf{1}\{c_x \leq 100 \log n\}$ by $\text{poly}(\log n)$ factors, so McDiarmid's Inequality inequality (applied to the appropriate Doob martingale) implies

$$\Pr\left(\left|\sum_x c_x^2 \cdot \mathbf{1}\{c_x \leq 100 \log n\} - \mathbb{E}\left[\sum_x c_x^2 \cdot \mathbf{1}\{c_x \leq 100 \log n\}\right]\right| \geq \frac{\varepsilon n}{200}\right) \leq e^{-\Omega\left(\frac{\varepsilon^2 n^2}{n \cdot \text{poly}(\log n)}\right)} \leq \frac{1}{n}$$

for sufficiently large $n$. Combining our two tail bounds with the triangle inequality and the union bound proves claim (3). $\qquad\square$

**Theorem 2.5.** *Suppose $n \geq \Omega(1/\varepsilon^2)$. Consider an algorithm which specifies a subset $T \subseteq [n]$, receives the set of points $\{x_i, i \in T\}$ and returns a $1 + \varepsilon/100$ approximation to $s(K)$, where $K$ is the underlying kernel matrix of the full set of $n$ points, with probability at least 99%. Then we must have $|T| \geq \Omega(\sqrt{n}/\varepsilon^2)$.*

*Proof.* Suppose we flip an unbiased coin and either give the algorithm points drawn from $\mathcal{D}(p_1)$ or $\mathcal{D}(p_2)$. If $n$ is sufficiently large, then from Lemma D.4, we know that with probability at least 99% that the kernel sum under case 1 is at least a $1 + \varepsilon/2$ factor larger than the kernel sum under case 2. Thus, an algorithm which returns a $1 + \varepsilon/100$ approximation can determine, with probability at least 99%, which distribution we are drawing from.

Let $x_1$ and $x_2$ be a sample from $\mathcal{D}(p_1)$ and $\mathcal{D}(p_2)$ respectively. Then the Hellinger distance, $d_H$, between $x_1$ and $x_2$ is bounded up to constant factors by

$$\sqrt{n} \cdot (\sqrt{1-p_1} - \sqrt{1-p_2})^2 + (\sqrt{p_1} - \sqrt{p_2})^2.$$

The first term can be bounded by

$$\left(\sqrt{\sqrt{n}-(1+\varepsilon)} - \sqrt{\sqrt{n}-1}\right)^2 = \left(\frac{\sqrt{n}-(1+\varepsilon) - \sqrt{n}+1}{\sqrt{\sqrt{n}-(1+\varepsilon)} + \sqrt{\sqrt{n}-1}}\right)^2 = O\left(\frac{\varepsilon^2}{\sqrt{n}}\right).$$

The second term is bounded by

$$\left(\sqrt{\frac{1+\varepsilon}{\sqrt{n}}} - \sqrt{\frac{1}{\sqrt{n}}}\right)^2 = O\left(\frac{\varepsilon^2}{\sqrt{n}}\right),$$

so

$$d_H(x_1, x_2) \le O\left(\frac{\varepsilon^2}{\sqrt{n}}\right).$$

Then by a standard relationship between total variation and Hellinger distance, we have that the total variation distance between $|T|$ samples from $\mathcal{D}(p_1)$ and $\mathcal{D}(p_2)$ is bounded by $O\left(\sqrt{|T|} \cdot \frac{\varepsilon}{n^{1/4}}\right)$. Thus if $|T| \ll \sqrt{n}/\varepsilon^2$, then the total varaition distance is bounded by $0.001$, contradicting the fact that we can distinguish the two distributions with probability $> 99\%$.

$\square$

## E. Towards a Lower Bound for Matrix-Vector Products for Mixed Sign Vectors?

The guarantees of Theorem A.1 naturally prompt us to ask if one can obtain such a guarantee for *arbitrary* vectors (with both positive and negative coordinates). Prior works which also studied matrix-vector products for kernel matrices ((Charikar & Siminelakis, 2017; Backurs et al., 2021; Indyk et al., 2025)) also require the input vector to be entry-wise non-negative to obtain a relative-error guarantee. Furthermore, it has been previously conjectured that the arbitrary vector case may require nearly quadratic time. Indeed, (Indyk et al., 2025) state that for this task "in general it may require computing $Kx$ exactly, which takes $\Omega(n^2)$ time". Unfortunately, the evidence given in (Indyk et al., 2025) is not formal[4]. More subtly, the evidence of (Indyk et al., 2025) actually *does not* require $\Omega(n^2)$ time, nor does it require computing $Kx$ using an $x$ with negative entries. In more detail, the argument of (Indyk et al., 2025) proceeds in two steps.

1. Having error $\varepsilon\|Kx\|_2$ actually implies $0$ error if it is the case that $Kx = 0$.

2. In general, computing $Kx$ exactly requires $\Omega(n^2)$ time.

There are issues with both steps of the reasoning. For (1), it may not always be the case that such an $x$ exists. Furthermore, finding such an $x$ could be computationally difficult itself. And lastly, if it is known that $Kx = 0$ beforehand, then computing $Kx$ is very easy, we can simply return $0$!

For step (2), the hard instance used in (Indyk et al., 2025) is a matrix of all zeros with a hidden $1$. This cannot happen in Kernel matrices since the diagonals are all $1$'s. Furthermore, detecting a $1$ is easy to do in linear time since it implies two vectors are identical (due to the fact that in all the kernels we study and all the natural kernels that we are aware of, $k(x, y) = 0 \iff x = y$). It *is* true that computing $Kx$ exactly in general should require $\Omega(n^2)$ time, but this also holds for non-negative $x$. Indeed, (Backurs et al., 2017) show that computing the sum of entries of $K$ *exactly* requires $\Omega(n^2)$ time assuming SETH, and the sum can be easily deduced if we can compute $K\mathbf{1}$ exactly.

Thus, there remains a big gap in our understanding of the complexity of matrix-vector products for kernel matrices. On one hand, we have near linear upper bounds for the non-negative case with strong relative-error grantees. On the other hand, virtually no non-trivial upper or lower bounds are known for the general case where the input vector has mixed signs.

Our aim is to bridge this gap and provide more evidence for the hardness of computing relative error estimates of $Kx$ for general $x$. This task seems out of reach of our techniques, but we show that a related intermediate problem requires nearly quadratic time, assuming SETH. To define the intermediate problem, we first define a related version of a Gaussian kernel matrix as follows. Given a dataset $X = \{x_1, \cdots, x_n\}$, define the $n \times (n+1)$ matrix $K'$ (we use the notation $K'$ as to not confuse with our original notion of a kernel matrix) as

- $K'_{i,i} = 1 \, \forall i$,

- $K'_{i,j} = e^{-\|x_i + x_j\|_2^2}$ for $1 \le i, j \le n$ with $i \ne j$,

- $K'_{i,n+1} = e^{-\|x_i\|_2^2}$.

---

[4]note that the authors are not claiming a formal statement, rather giving a heuristic derivation, so the following discussion does not contradict any of their formal theorems in the paper

Some explanation is in order. $K'$ is like a Gaussian kernel matrix in many ways: it's main diagonal is all $1$'s. For the other entries, instead of using the kernel function $e^{-\|x-y\|_2^2}$, we use the kernel function $e^{-\|x+y\|_2^2}$. Lastly, the matrix is not a square matrix and there is one extra column for the all zeros vector. That is, $K'$ is "asymmetric" up to only *one vector*: the rows are indexed by $X$ and the columns are indexed by $X \cup \{0\}$.

We note that for $K'$ it is easy to output a relative error matrix-vector product for non-negative vectors $x$: that is, given the dataset $X$ and a vector $x \in \mathbb{R}^{n+1}$, we can (in the same running time as Theorem 2.1) output $y = K'x + e$ satisfying $\|e\|_2 \le \varepsilon \|K'x\|_2$: indeed, we can calculate the contribution of the last column exactly in $O(n)$ time. For the "square" part of $K'$, one can easily check that the same algorithm of the general theorem A.1 works, since it reduces the MVP computation to KDE queries, which we can also instantiate for the kernel $e^{-\|x+y\|_2^2}$ by simplify multiplying dataset points by $-1$ for which the KDE data structure is constructed.

Thus, this family of matrices (where we use the Gaussian kernel with a $+$ sign), we can obtain strong relative error guarantees. Our main theorem below shows that this is not the case if we instead query $K'$ with a vector $x$ that may have mixed signs.

**Theorem 2.6.** *Consider the matrix $K'$ defined above. Any algorithm which takes as input dataset $X$, vector $x \in \mathbb{R}^{n+1}$, and returns a $y$ such that $y = K'x + e$ with $\|e\|_2 \le n^{-0.002} \cdot \|K'x\|_2$ requires almost quadratic time, assuming SETH.*

To prove the theorem, we use reduction from the Orthogonal Vectors (OV) problem, showing that the OV problem can be solved if such a hypothetical algorithm of Theorem 2.6 exists. First, the following lemma helps transform the input of OV (Definition 3.1) into a more convenient form.

**Proposition E.1.** *Let $(A, B)$ with $A = \{a_1, \cdots, a_n\}$ and $B = \{b_1, \cdots, b_n\}$ be the input to the Orthogonal Vectors problem in dimension $d = \omega(\log n)$. By increasing the dimension $d$ by a constant multiplicative factor, we can ensure the following:*

- *$\forall i, j \in [n], \langle a_i, a_j \rangle \ne 0$ and $\langle b_i, b_j \rangle \ne 0$; that is, none of the pairs within $A$ and within $B$ are orthogonal.*

- *$\forall i, j \in [n], \|a_i\|_1 = \|b_j\|_1$, that is, all vectors have the same number of $1$'s.*

*Proof.* We pad the vectors with $2d + 2$ extra coordinates as follows: for the first extra coordinate, all vectors in $A$ will get a $1$ and all vectors in $B$ will get a $0$. Similarly in the second extra coordinate, all vectors in $B$ will get a $1$ and all vectors in $A$ will get a $0$. For the next block of $d$ coordinates, all vectors in $B$ get all zeros. For vectors $a \in A$, we give them $d - \|a\|_1$ many $1$'s (say consecutive), and the rest zeros (where we use the original number of 1s). We do the symmetric operation on the second block of $d$ coordinates for vectors in $B$. This ensures the two stated guarantees □

We now show that a sub-quadratic algorithm for computing matrix vector products for arbitrary vectors for the matrix $K'$ allows us to solve OV.

*Proof of Theorem 2.6.* Let $X = A \cup B$ where $A, B$ are the input point sets for the OV problem, with the transformation specified in Proposition E.1 applied to $A$ and $B$. Let $t$ be the number of $\pm 1$ coordinates in the vectors in $A \cup B$ (this quantity is the same for all vectors in $X$ due to Proposition E.1). First, we can assume without loss of generality that there are at most $n^{0.001}$ pairs $a \in A, b \in B$ that satisfy $\langle a, b \rangle = 0$, since otherwise by random sampling, we can find such a pair in $n^{1.99999} = o(n^2)$ time with high probability. By scaling the points by a factor $\sqrt{C \log n}$ for a sufficiently large constant $C \ge 1$, we have the following:

1. For $a \in A$ and $b \in B$, if $\langle a, b \rangle = 0$, then the corresponding entry in $K'$ has value (note the plus)

$$e^{-C \log(n) \|a+b\|_2^2} = e^{-2C \log(n) t} = n^{-2Ct}.$$

2. On the other hand, if $\langle a, b \rangle \ge 1$, the corresponding entry in $K'$ has value

$$e^{-C \log(n) \|a+b\|_2^2} \le e^{-C \log(n)(2t+2)} \le n^{-(2t+2)C}.$$

3. Furthermore, the $n + 1$th column of $K$ is a constant vector since $\|x_i - 0\|_2^2 = \|x_i\|_2^2 = \|x\|_1$ is the same value for all $x_i \in X$ due to Proposition E.1. Let $\Delta = e^{-C \log(n) t}$ be the constant value in the $n + 1$th column.

4. Finally for every pair $a \neq a' \in A$, the corresponding entry in $K'$ has value

$$e^{-C \log(n) \|a+a'\|_2^2} \leq e^{-C \log(n)(2t+2)} \leq n^{-(2t+2)C},$$

and a similar calculation holds for a pair $b \neq b' \in B$.

Consider the vector

$$x = (1, \cdots, 1, -1/\Delta) \in \mathbb{R}^{n+1}. \tag{9}$$

For $i \in [n]$, the $i$th entry of $K'x$ has value

$$\sum_{j=1}^n e^{-C \log(n)\|x_i+x_j\|_2^2} + \Delta \cdot \frac{-1}{\Delta} = \sum_{j=1, j \neq i}^n e^{-C \log(n)\|x_i+x_j\|_2^2}.$$

In other words, the $i$th entry of $K'x$ has value equal to the sum of the $i$th row of $K'$, ignoring the diagonal (note: removing the diagonal is the main hurdle and the extra coordinate allows us to do this).

Now in the case that an orthogonal pair exists, item (1) above implies that some entry of $K'x$ has value at least $n^{-2Ct}$. In the case where no such pair exists, item (2) implies that every entry of $K'x$ is bounded by $n^{-(2t+1)C}$. Furthermore, by our assumption that there are at most $n^{0.001}$ orthogonal pairs in the yes case, we know that in *both cases*, we can upper bound $\|K'x\|_2^2$ by

$$\|K'x\|_2^2 \leq n^{0.001} \cdot (n^{0.001} n^{-2Ct})^2 + n \cdot (n^{-(2t+2)C})^2 \leq O(n^{-4Ct+0.003}).$$

This implies that *every* entry of $e$ is upper bounded by $\varepsilon \|K'x\|_2 = O(\varepsilon \cdot n^{-2Ct+0.0015})$. Thus if $\varepsilon \leq n^{-0.0015}/C'$ for a sufficiently large constant $C' \geq 1$, then in the yes case, some coordinate of $K'x + e$ has value at least $n^{-2Ct}/100$. Conversely in the no case, all coordinates of $K'x + e$ are smaller than $n^{-(2t+1)C} + \varepsilon \|K'x\|_2 \leq n^{-2Ct}/1000$. Thus, computing $y = K'x + e$ satisfying $\|e\|_2 \leq \varepsilon \|K'x\|_2$ allows us to solve OV, and the conclusion follows. $\square$

### E.1. Extension to Vector-Matrix-Vector Products

We extend Theorem 2.6 to the case of computing $y^\top K'x$ for the same matrix $K'$ as above.

**Theorem E.2.** *Consider the matrix $K'$ described above. Any algorithm which takes as input $X$, vectors $x \in \mathbb{R}^{n+1}$ and $y \in \mathbb{R}^n$ such that the first $n$ coordinates of $x$ and $y$ agree, and returns a $n^{O(1)}$ multiplicative approximation to $y^\top K'x$ requires almost quadratic time, assuming SETH.*

*Proof.* We use a similar reduction from OV as in the proof of Theorem 2.6. Recall from the proof of Theorem 2.6 the definition of vector $x$ (Eq. (9)) and the construction of the kernel matrix $K' \in \mathbb{R}^{n \times n+1}$. Let $y$ be the all ones vector.

From the proof of Theorem 2.6, we know that the $i$th entry of $K'x$ is equal to the sum of the $i$th row of $K'$, ignoring the diagonal. Thus, $y^\top K'x$ denotes the sum of the off diagonal entries of $K$.

Now continuing as in the proof of Theorem 2.6, item (1) there implies that some entry of $K'x$ has value at least $n^{-2Ct}$ if an orthogonal pair exists, where $t$ is the (fixed) number of 1 coordinates in $A \cup B$. In the case where no such pair exists, item (2) there implies that every entry of $K'x$ is bounded by $n^{-(2t+1)C}$. In other words, if an orthogonal pair exists, then $y^\top K'x$ is at least $n^{-2Ct}$, where as if no such pair exists, $y^\top Kx \leq n^{-(2t+1)C+1} = n^{-2tC} \cdot n^{-C+1} \ll n^{-2Ct}$ by picking a large constant $C \geq 1$. Thus, any polynomial approximation to $y^\top K'x$ allows us to solve OV, proving the lower bound. $\square$

## F. Lower Bounds for Kernel Sums

In this section, we prove lower bounds for calculating arbitrary vector-matrix-vector products, as well as computing on *asymmetric* kernel matrices, where the rows and columns need not be indexed by the same point set.

**Theorem 2.7.** *For any $d = \omega(\log n)$, computing $v^\top Kw$ for non-negative $v, w$ for the Gaussian kernel matrix $K \in \mathbb{R}^{n \times n}$ up to polynomial relative error requires almost quadratic time, assuming SETH.*

*Proof.* Let $A, B$ be the input to OV. Note that without loss of generality, we can assume every vector in $A$ and $B$ have the same number of 1's (say $t$). This is because we can pad $2d$ extra entries to the vectors, set the first $d$ new coordinates to all

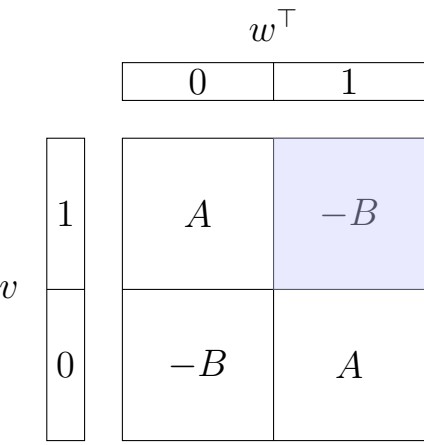

*Figure 3.* Representation of $v^\top K w$.

zeros for all vectors $A$ and the second set of $d$ to all zeros for all vectors in $B$. Then for vectors in $A$, we set the appropriate number of the second set of $d$ coordinates to 1's and vice-versa for $B$.

Now let $X = A \cup -B$ be the underlying dataset for the kernel matrix $K$ with kernel function $k(x,y) = e^{-C \log(n)\|x-y\|_2^2}$, let $v$ be the vector with all 1's in the first half and zeros in the second half, and $w$ be the vector with all 1's in the second half and zeros in the first half.

For $a \in A$ and $b \in B$, if $\langle a, b \rangle = 0$, then the corresponding entry in $K$ has value

$$e^{-C \log(n)\|a+b\|_2^2} = e^{-2C \log(n)t}.$$

On the other hand, if $\langle a, b \rangle \geq 1$, the corresponding entry in $K$ has value

$$e^{-C \log(n)\|a+b\|_2^2} \leq e^{-C \log(n)(2t+2)}.$$

Thus, by our choice of $v$ and $w$ (see Figure 3), we have that $v^\top K w$ is exactly the sum of all the entries

$$S := \sum_{a \in A, b \in B} e^{-C \log(n)\|a+b\|_2^2} = e^{-C \log(n)\cdot 2t} \sum_{a \in A, b \in B} e^{-2C \log(n)\langle a,b\rangle}.$$

If there is no orthogonal pair, then this sum is at most $e^{-C \log(n)\cdot(2t+2)} \cdot n^2$ and if there is an orthogonal pair, then this sum is at least $e^{-C \log(n)\cdot(2t+2)} \cdot (n^2 - 1) + e^{-2C \log(n)t}$. The ratio of these values is at least

$$\frac{e^{-C \log(n)\cdot(2t+2)} \cdot (n^2 - 1) + e^{-2C \log(n)t}}{e^{-C \log(n)\cdot(2t+2)} \cdot n^2} = \frac{n^{-2C+2} - n^{-2C} + 1}{n^{-2C+2}} \geq \Omega(n^{2C-2}).$$

By setting $C$ to be a sufficiently large constant, we have that any $o(n^{2C-2})$ approximation to $v^\top K w$ decides if we have an orthogonal pair in $A, B$, solving the OV problem.

$\square$

Ideas of the above lower bound can be extended to show hardness for computations on *asymmetric* kernel matrices. Note that in the introduction, we defined a kernel matrix to be symmetric where the rows and columns are indexed by the same point set $X$. In practice, asymmetric kernel matrices are also popular, where an asymmetric kernel matrix $K \in \mathbb{R}^{n \times n}$ has its rows and columns indexed by possibly different point sets $X$ and $Y$: $K_{i,j} = k(x_i, y_j)$.

**Corollary F.1.** *For any $d = \omega(\log n)$, computing $\mathbf{1}^\top K \mathbf{1}$ for an asymmetric kernel matrix $K \in \mathbb{R}^{n \times n}$ up to polynomial relative error requires almost quadratic time, assuming SETH.*

*Proof.* The proof uses the same construction as in Theorem 2.7: let $A, B$ be the input to OV and consider the 'asymmetric' kernel matrix $K_{A,-B}$. If we let the kernel function be $k(x,y) = e^{-C \log(n)\|x-y\|_2^2}$ then the same calculation as in

Theorem 2.7 implies that the sum of the entries of $K$ is

$$\sum_{a \in A, b \in B} e^{-C \log(n) \|a+b\|_2^2} = e^{-C \log(n) \cdot 2t} \sum_{a \in A, b \in B} e^{-2C \log(n) \langle a, b \rangle}.$$

Thus to repeat the argument, if there is no orthogonal pair, then this sum is at most $e^{-C \log(n) \cdot (2t+2)} \cdot n^2$ and if there is an orthogonal pair, then this sum is at least $e^{-C \log(n) \cdot (2t+2)} \cdot (n^2 - 1) + e^{-2C \log(n)t}$. The ratio of these values is at least $\Omega(n^{2C-2})$ as before, and by setting $C$ to be a sufficiently large constant, we have that any $o(n^{2C-2})$ approximation to $\mathbf{1}^\top K \mathbf{1}$ decides if we have an orthogonal pair in $A, B$, solving the OV problem. $\square$

Theorem 2.7 also implies that approximating $\lambda_1$ for an asymmetric kernel matrix requires almost quadratic time, assuming SETH.

**Corollary F.2.** *For any $d = \omega(\log n)$, computing the top singular value of $K$ for an asymmetric kernel matrix $K$ up to polynomial relative error requires almost quadratic time, assuming SETH.*

*Proof.* We use the same construction as in Theorem 2.7. Note that the frobenius norm squared of the matrix $K$ is

$$\sum_{a \in A, b \in B} e^{-2C \log(n) \|a+b\|_2^2} = e^{-2C \log(n) \cdot 2t} \sum_{a \in A, b \in B} e^{-4C \log(n) \langle a, b \rangle}.$$

If there is no orthogonal pair, then the sum is at most $n^2 e^{-4C \log(n) \cdot (t+1)}$, meaning the top singular value is at most $n \cdot e^{-2C \log(n) \cdot (t+1)} \leq n^{-2Ct - 2C + 1}$. On the other hand, an orthogonal pair implies an entry with value at least $e^{-2C \log(n) \cdot t} = n^{-2Ct}$, which lower bounds the top singular value, e.g. by considering the basis vector corresponding to the column that this entry is in. Thus, any algorithm outputting a fixed polynomial factor approximation solves OV by appropriately increasing the value of the constant $C$. $\square$

**Corollary F.3.** *For any $d = \omega(\log n)$, computing $K\mathbf{1}$ for an asymmetric kernel matrix $K$ up to polynomial relative error requires almost quadratic time, assuming SETH.*

*Proof.* Let $v$ be the computed value of $K\mathbf{1}$, which we suppose for the sake of contradiction can be expressed as $v = K\mathbf{1} + n^\alpha \|K\mathbf{1}\| u$ for some non-negative unit vector $u$. The proof uses the same construction as in Theorem 2.7. Note that the $i$th entry of $K\mathbf{1}$ is

$$\sum_{b \in B} e^{-2C \log(n) \|a_i + b\|_2^2} = e^{-2C \log(n) 2t} \sum_{b \in B} e^{-4C \log(n) \langle a_i, b \rangle}.$$

If there are no orthogonal pairs, then

$$\|K\mathbf{1}\| \leq \sqrt{n} e^{-2C \log(n)(2t+2)}$$

so

$$\|v\| \leq (1 + n^\alpha) \|K\mathbf{1}\| \leq e^{-2C \log(n)(2t+2) + (\alpha + \frac{1}{2}) \log(n) + 1}.$$

On the other hand, if there is an orthogonal pair (say $a_i$ and $b_j$), then

$$(K\mathbf{1})_i = e^{-2C \log(n) 2t} \sum_{b \in B} e^{-4C \log(n) \langle a_i, b \rangle} \geq e^{-2C \log(n) 2t}$$

so

$$\|v\| \geq e^{-2C \log(n) 2t}.$$

When $C$ is large enough, this allows one to distinguish between the two cases. $\square$

# G. Empirical Results

While the main focus of our paper is on proving improved theoretical bounds, we also empirically evaluate our algorithm for approximating the top eigenvalue $\lambda_1$ of the underlying (symmetric) kernel matrix $K$ (Algorithm 2). It has strong theoretical guarantees, obtaining $\varepsilon$ relative error in sub-quadratic time. Its analysis, given in Appendix C, is also subtle, even though Algorithm 2 is itself quite simple, so it makes a perfect case to study the interplay of theory and practice.

Our main goal is to show that our choice of the error of the noisy matrix vector product (see Definition 2.1), namely $\Theta(\varepsilon)$ in Theorem A.3, *is sufficient* in practice to obtain a $\Theta(\varepsilon)$-relative error in the approximation of $\lambda_1$. This is already proven in Appendix C, and here we demonstrate that the phenomenon can also be observed in experiments. Thus, we demonstrate that the parameter choices of prior work of (Backurs et al., 2021), which uses an MVP with approximation $\Theta(\varepsilon^2)$ to get $\Theta(\varepsilon)$ relative error for $\lambda_1$, *is unnecessary*. This is important to demonstrate empirically for kernel matrices because an $\Theta(\varepsilon^2)$ approximate MVP algorithm is much slower than than an $\Theta(\varepsilon)$-approximate MVP, since KDE data structures (which are repeatedly called in an approximate MVP) already have an underlying $1/\varepsilon^2$ scaling, meaning a quadratic blow up in the accuracy of the MVP can translate into a large overhead in the running time.

Indeed, our improved parameter choice (with the matching analysis) is a main contributor to the $\text{poly}(1/\varepsilon)$ decrease in running time that our Theorem A.3 obtains over the prior state of the art of (Backurs et al., 2021). Even a modest $\varepsilon$ value of say $\varepsilon = 0.01$ can lead to an unnecessary huge overhead of orders of magnitude in the running time, if one closely follows the theoretical guidelines of (Backurs et al., 2021) in practice. We refer to Section 4.2 for further discussions of the choice of the approximation error in the MVP and its impact in the error the top eigenvalue estimate. We also remark that for large $n$, exact eigenvalue computation requires at least $\Omega(n^2)$ time, which is prohibitive, necessitating the need for a fast approximation.

## G.1. Empirical Setup

We use the Exponential Kernel $k(x, y) = \exp(-\|x - y\|_2/\sigma)$ for our main experiments. This means we use the same approximate MVP algorithm of (Backurs et al., 2021), allowing us to study the effect of matrix-vector products on the relative error of the $\lambda_1$ estimate in isolation. We pick the bandwidth parameter $\sigma$ so that the average entry in the kernel matrix is approximately $10^{-3}$. This is a standard choice in experiments with KDE data structures (Karppa et al., 2022), and other choices produced similar results.

**Implementation Details.**    The phenomenon that we are studying (how does the approximation quality of matrix vector products affect the relative error of the $\lambda_1$ estimate) is agnostic to how exactly the approximate matrix vector products are implemented. We pick the most practically convenient method as detailed below. First, we describe our choice of the KDE algorithm. Note that our algorithms use KDE queries as a black box. Thus they are valid for any kernel admitting KDE data structures. We use random sampling as the underlying KDE datastructure for implementing the approximate non-negative matrix-vector product algorithm (described below). It can be checked that random sampling satisfies Definition A.1 with $p = 1$. While the data structures of Table 3 offer better theoretical guarantees, the state of the art Exponential Kernel KDE algorithm has not been implemented to our knowledge. On the other hand, random sampling is trivial to implement and is very fast in practice, as affirmed by a recent empirical study of (Karppa et al., 2022). Note that (Karppa et al., 2022) also proposes another practical algorithm, called DEANN, which is competitive with random sampling. We chose not to use it in our final experiences since DEANN was still slower than random sampling in Python (even with Numba optimization). It also requires pre-building a nearest neighbor index, whereas random sampling does not require any preprocessing. However, we again emphasize that our algorithms has the advantage that any progress on KDE datastructures (whether in theory or practice) automatically translates into faster algorithms since KDE queries are used in a black box manner. For random sampling, the main parameter is how many points to sample when given $\varepsilon$ as input in Definition A.1. We set this to $\lceil \frac{1}{\varepsilon^2} \rceil$ to align with the theoretical guarantees.

Lastly to implement the $\varepsilon$-approximate non-negative matrix-vector product for the Exponential Kernel, we use a practically optimized version of the algorithm of (Backurs et al., 2021): we partition the coordinates of the input (non-negative) unit vector into groups that only differ by $1 + \varepsilon$. For every coordinate of the output vector, the contribution of every group is just a KDE query, which we use random sampling for as above.

We implement all algorithms using Python 3.9.7 on an M1 MacbookPro with 32GB of RAM. We use Numba to accelerate all the numerical Python code. In all experiments we repeat independently at least 20 times and shade one standard deviation.

**Metrics.** We measure accuracy by the relative error. For approximating top eigenvalue using a corresponding vector $v$, the relative error is $1 - v^T K v / \lambda_1(K)$. $\lambda_1$ is computed by running the standard exact power method run until convergence. We also measure the accuracy of a matrix vector product approximation $u$ of $Kv$ using $\|u - Kv\|_2 / \|Kv\|_2$. This value corresponds to how the error is defined in Definition 2.1.

**Datasets.** $n$ denotes the number of data points, but we note that the underlying kernel matrix is of size $n^2$. We use datasets up to size $n = 10^4$ because we were not able to compute the exact matrix vector product or the exact $\lambda_1$ for larger datasets, since the exact computation scales quadratically. We use three real world datasets that have been previously used for evaluating KDE based algorithms: The Forest CoverType dataset (Blackard & Dean, 1999), used in (Siminelakis et al., 2019; Backurs et al., 2019; 2021), MNIST, used in (Backurs et al., 2021), and CLIP embeddings of CIFAR10, used in (Backurs et al., 2024). For MNIST we select $n = 1000$ random data points, which are in dimension 784. For CoverType, we select $n = 10^4$ random data points, which are in dimension 54, and for CLIP embeddings we select 500 points in dimension 512 in our main experiments. Later we also select larger subsets of these datasets for wall-clock studies.

### G.2. Main Results

We vary the $\varepsilon$ approximation factor used in the matrix vector product algorithm and measure the accuracy of the output, which is an approximation to $\lambda_1$ in the power method of Algorithm 2. We set the number of iterations to a large value so that the power method has converged (in all of our experiments, we noticed that around 50 iterations is ample for convergence). Theorem A.3 predicts that the relationship between the relative error of the approximation and the error of the noisy matrix vector product should be linear, and indeed, we observe a linear relationship in all of our datasets as shown in Fig. 4. This empirically corroborates our theoretical analysis: **using approximate matrix vector products of accuracy $\Theta(\varepsilon)$ is the correct choice** if we want an $\varepsilon$ relative error approximation of $\lambda_1$. This also empirically demonstrates that the prior analysis of (Backurs et al., 2021) is loose as it instead predicts a square-root relationship: their analysis initializes approximate matrix vector products with the guarantee that for every non-negative $x$, they output a $y$ with $\|y - Kx\|_2 \leq \varepsilon^2 \|Kx\|_2$, to obtain a final relative error of $\Theta(\varepsilon)$ for $\lambda_1$. Thus, Fig. 4 confirms that our tighter bounds translates to actual computational savings via tighter instantiation of parameters.

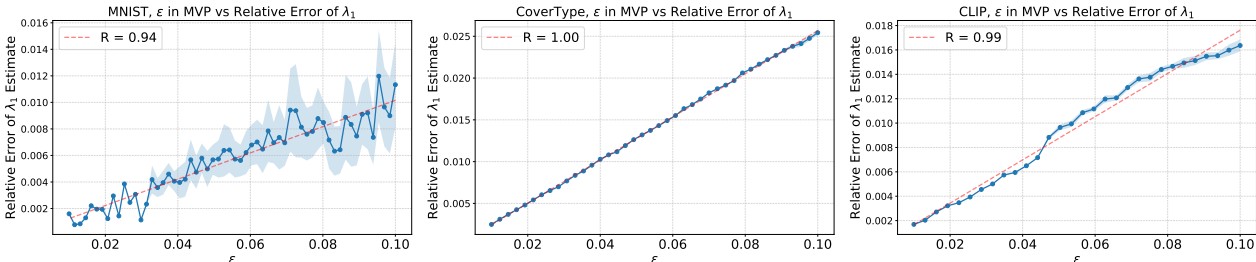

*Figure 4.* Plotting the relative error of the final $\lambda_1$ estimate, against the error $\varepsilon$ of the matrix vector products used in Algorithm 2. The datasets are MNIST, CoverType, and CLIP embeddings from left to right.

A natural follow up question is if one can get *even better* than linear scaling between the approximation used in the MVP step and the final relative error of $\lambda_1$ (i.e. if $\Theta(\varepsilon)$-approximate matrix vector products can lead to $o(\varepsilon)$ relative error). The only parameter that could potentially influence this is the number of iterations taken in the power method. Our result of Proposition C.3 already proves that this cannot happen in the worst case, and indeed, we empirically observe this phenomenon in practice as well. Fig. 5 shows that even if we let the number of iterations become very large, the noise of the approximate matrix vector product dictates the final error of the relative error of $\lambda_1$ and sets a fundamental lower bound of how accurate the final error can be, even if we run power method for a very large number of iterations. Note that if one were to use exact matrix vector products (which require $\Omega(n^2)$ time, the error monotonically decreases. In the CLIP figures (third row), the very first iteration achieves around $0.04$ relative error because the starting vector of all ones already happens to obtain $0.04$ relative error. However, further iterations are required with sufficiently accurate matrix vector products to obtain even smaller relative error.

Lastly, we briefly note that the choice of the scale parameter does not affect our theoretical analysis and indeed, across other

choices of bandwidth, we observe the same linear scaling as in Fig. 4. We varied the bandwidth so that the average kernel matrix value was now $0.01$, which is another common choice in practice (Karppa et al., 2022), and observed qualitatively similar results as in Fig. 4 in Fig. 6.

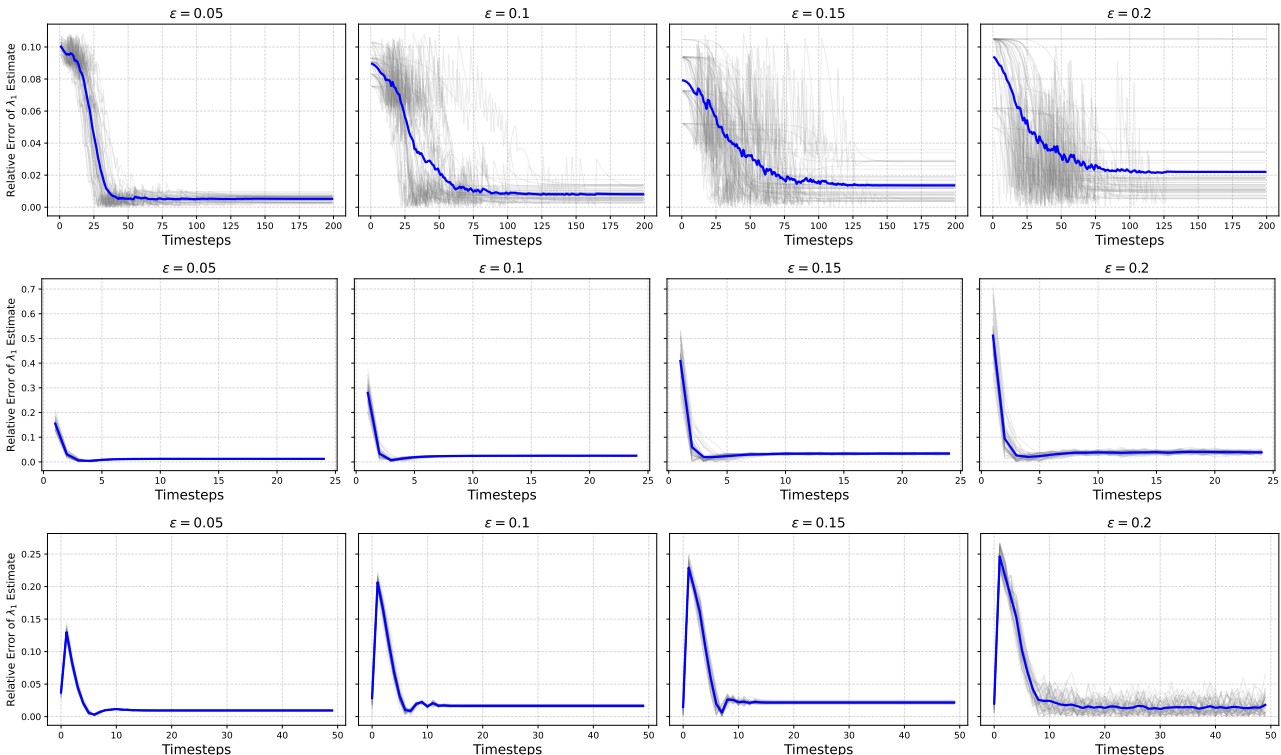

*Figure 5.* Trajectories of the noisy power method (Algorithm 2) as we vary the number of power method iterations, as a function of the approximation of the matrix vector product. As the number of iterations grows, the final relative error of the top eigenvalue approximation converges to a value depending on the matrix vector product approximation. The top row corresponds to the MNIST dataset, the middle row corresponds to CoverType, and the last row is for CLIP embeddings.

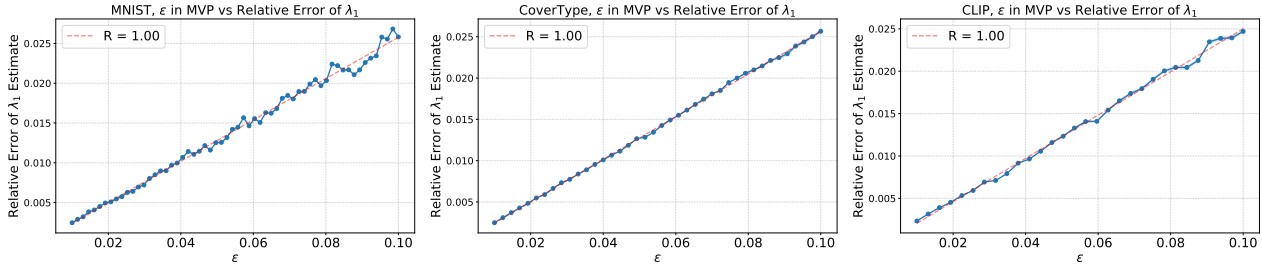

*Figure 6.* An analogous plot to Fig. 4, except we change the bandwidth parameter so that the average entry of the kernel matrix is $\approx 0.01$. The linear relationship of Fig. 4 is maintained.

### G.3. Comparison to Additive Error Methods

In our theoretical study and experiments, we focus on algorithms which obtain *relative error* approximation to $\lambda_1$. That is, we require the unit vector $u$ returned by the algorithm to satisfy

$$u^\top K u \geq (1 - \varepsilon)\lambda_1(K).$$

In comparison, algorithms which obtain additive error, e.g. those based on row or column sub-sampling, give the weaker guarantee of approximating the top eigenvalue up to additive factor, e.g. the sampling algorithm of (Swartworth & Woodruff, 2025) gives a $O(\varepsilon n)$ additive factor by sampling a $1/\varepsilon \times 1/\varepsilon$ sub-matrix. Additive error guarantees of this form are strictly

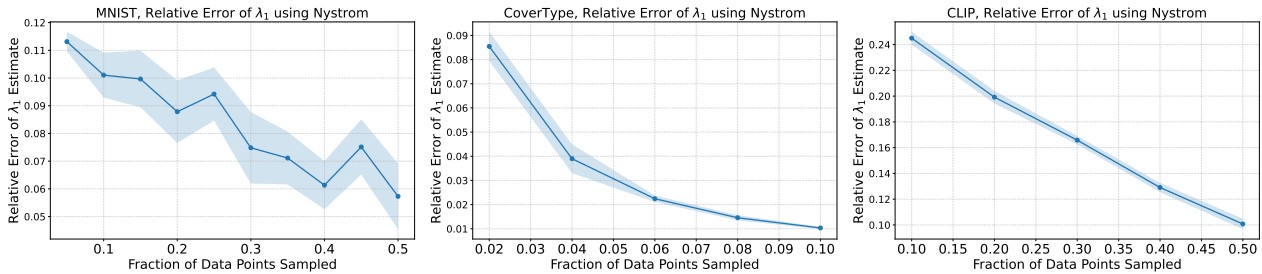

*Figure 7.* Relative error of approximating $\lambda_1$ as a function of number of samples taken by the Nystrom method. We can observe that a large constant fraction of the data points needs to be sampled to get relative error less than 0.1.

weaker than relative error since $\lambda_1 \leq n$ for any real symmetric matrix with entries bounded by 1 in absolute value, which is the case for our kernel matrices. The family of row or column sampling algorithms also includes the well studied Nystrom method (Gittens & Mahoney, 2016). We perform further experiments on our datasets (on the exponential kernel with the same scale as in Fig. 4) to show that the Nystrom method is not sufficient for obtaining $\varepsilon$-relative error, unless a prohibitive number ($\approx$ linear) of points are sampled. We vary the number of data points sampled by the Nystrom method (we use a standard scikit-learn implementation (Pedregosa et al., 2011)) and calculate the top eigenvalue of the resulting Nystrom approximation. We then measure the relative error of the top eigenvalue of the Nystrom approximation compared to $\lambda_1$ of $K$, the true kernel matrix.

The results are shown in Fig. 7, where we see that a large constant factor of the entire dataset must be sampled by Nystrom to get small relative error; e.g for MNIST we are sampling half of the dataset to get 0.05 relative error. However, in the case that $\Omega(n)$ data points are sampled, this offers no asymptotic advantage over the naive quadratic time algorithm of just instantiating the kernel matrix exactly. Thus we believe our result highlights that row or column sub-sampling based methods such as Nystrom may not be well-suited for high-precision approximations of $\lambda_1$,[5] unlike our version of the power method of Algorithm 1 (instantiated with approximate matrix vector products), which allow for better relative error as one increases the accuracy of the noisy matrix vector products (see Fig. 4). This also directly has a direct measurable impact on the running time. For example, in the MNIST dataset, using $\varepsilon = 0.1$ as input in the noisy matrix vector product and then running for only 10 iterations already gives smaller than 0.03 relative error. This took on average $455ms$. On the other hand, Nystrom with sampling 50% of the data points is only able to give 0.05 relative error, but takes more than 30 seconds, which is more than one magnitude slowdown, for a worse approximation. Note that the reason row or column sampling methods struggle with high accuracy is that their additive error typically scales with large factors such as $n$ (in the guarantees of (Swartworth & Woodruff, 2025)) or $\|K\|_F$ (in guarantees such as sampling by row norms (Woodruff et al., 2014)). Driving down the additive error factor to be comparable to relative error thus requires a large number of samples.

### G.4. When Should One Use Approximate Matrix-Vector Products?

We now discuss when it is appropriate to use an exact matrix vector product for non-negative vectors over using our practical approx. MVP algorithm for the Exponential Kernel. Our focus will be on the wall-clock running time in our specific hardware (see above). Note that theoretically, the approximation algorithms for MVPs are much faster and have a sub-quadratic scaling in $n$, e.g. for the exponential kernel these algorithm have a $n^{1.1}$ dependency (see Table 3). On the other hand, an exact matrix vector product takes $\Omega(n^2)$ time. Since these algorithms gives an asymptotic improvement, we wish to better understand when the 'asymptotics kick in' in the real world datasets of our experiments. We note that a similar study was conducted in (Backurs et al., 2021), but no actual running times were given there. Instead, (Backurs et al., 2021) focused on number of kernel evaluations performed by exact versus approximate methods. Our goal is to extend their study and provide meaningful parameter ranges when one can expect wall-clock speedups.

Clearly, the answer is a function of the dataset size $(n, d)$ and the desired approximation factor $\varepsilon$. Keeping $\varepsilon$ fixed, as $n$ increases then sub-quadratic methods become comparatively much faster than the naive quadratic time method. Fixing $n$, as $\varepsilon$ decreases, these algorithms have a polynomial in $1/\varepsilon$ dependency, so they become slower as we compute a high precision answer.

We can observe these trends in practice. In the MNIST dataset (see the top row of Fig. 8), we plot the running time of our

---

[5]These methods are of course very useful for many other downstream tasks; see (Williams & Seeger, 2000).

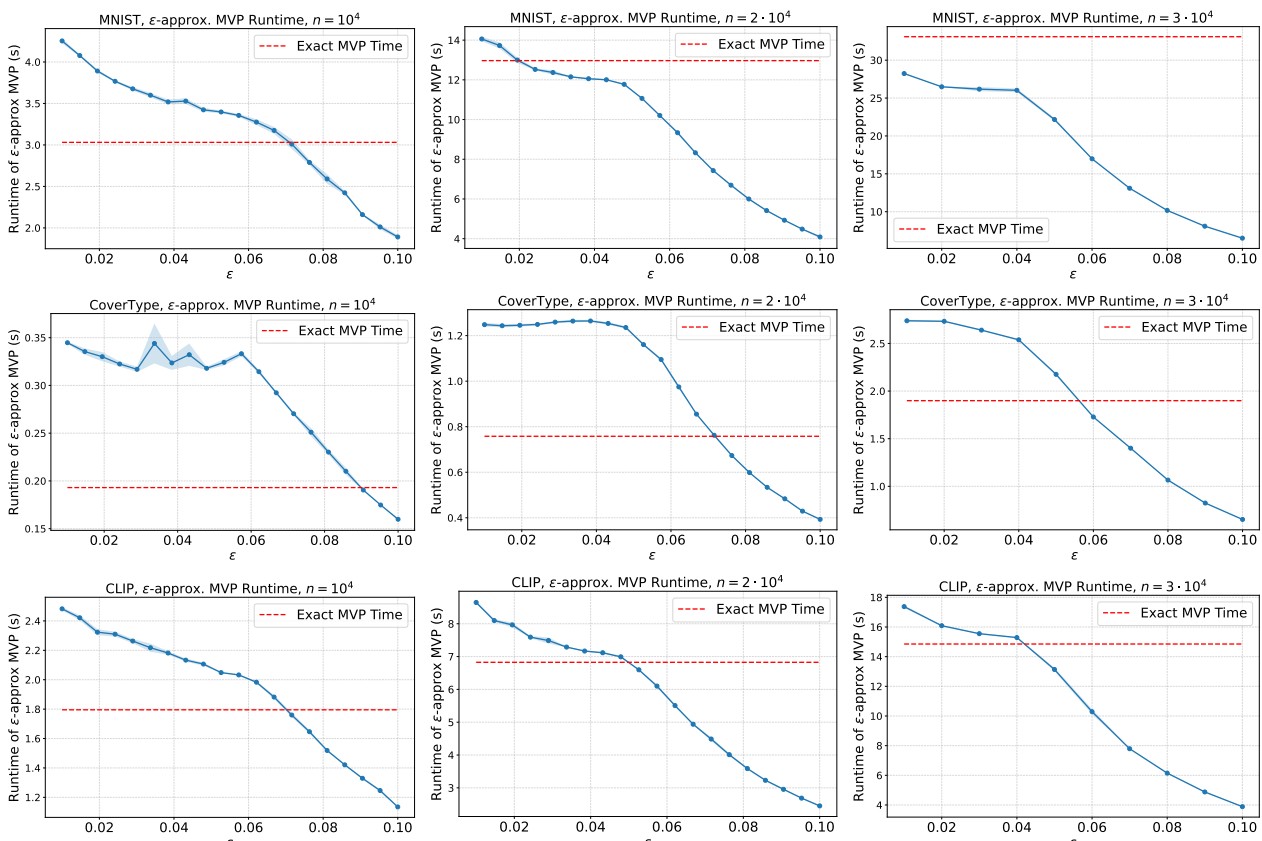

*Figure 8.* We vary the dataset sizes as well as $\varepsilon$ in the approximate matrix vector computation, and plot running time compared to an exact computation.

practical $\varepsilon$-approximate matrix vector product as a function of $\varepsilon$, as well as the running time of an exact computation, shown as a dashed red line. We do this for three increasing point sets: $n = 10^4, 2 \cdot 10^4$, and $3 \cdot 10^4$. Clearly, a smaller $\varepsilon$ requires a longer running time for a single matrix-vector computation. As the dataset size grows (left to right), the 'cut-off' point in our implemented $\varepsilon$-approximate matrix-vector product becomes more favorable. Indeed, when $n = 3 \cdot 10^4$, any $\varepsilon \geq 0.01$ is faster than the exact computation. Furthermore, the approximate computations become *increasingly* faster: for $n = 10^4$, an $\varepsilon = 0.1$ computation is approximately 1.5x faster than an exact computation. But for $n = 3 \cdot 10^4$, it is $> 4$x faster. We observe qualitatively similar trends in our other two datasets as well.

The results of Fig. 8 thus lead to the (expected) conclusions that smaller $\varepsilon$ makes an exact matrix vector computation more favorable, whereas a larger $n$ makes an exact computation less favorable, with the 'cutoff' in our experiments being around $n \approx 10^4$. One may wonder then how one should choose the right $\varepsilon$. Unfortunately, but expectedly, the answer here is less universal and depends on the desired application at hand. However, it is true that using a smaller $\varepsilon$ in our approximate mat-vec implementation leads to more accurate downstream result: indeed Fig. 9 demonstrates that across our three datasets, smaller values of $\varepsilon$ lead to more accurate computation of the top eigenvalue (all using 75 power method iterations), even if we vary the scale parameter of the kernel (which is represented as the average kernel value on the x-axis). For our datasets and for our task of top eigenvector computation, with setting $\varepsilon = 0.1$ with a resulting $\approx 0.03$ relative error for our $\lambda_1$ approximation seems to be a reasonable choice. This would be more than a 3x speedup in all of our datasets for $n = 3 \cdot 10^4$ compared to exact computation, if we fix the number of power method iterations. But again we emphasize that the right choice of $\varepsilon$ will depend on the downstream application at hand, and a larger value of the dataset, $n$, is needed to justify using a smaller $\varepsilon$ in practice, over exact computation.

Note that other considerations as the type of hardware (e.g. running on a GPU) may affect the specifics of these results, but we envision the *trends* to stay consistent; we leave this as an interesting direction for future work. At larger $n$, other KDE algorithms in practice may also be preferable, such as DEANN; this is also another interesting direction for future work.

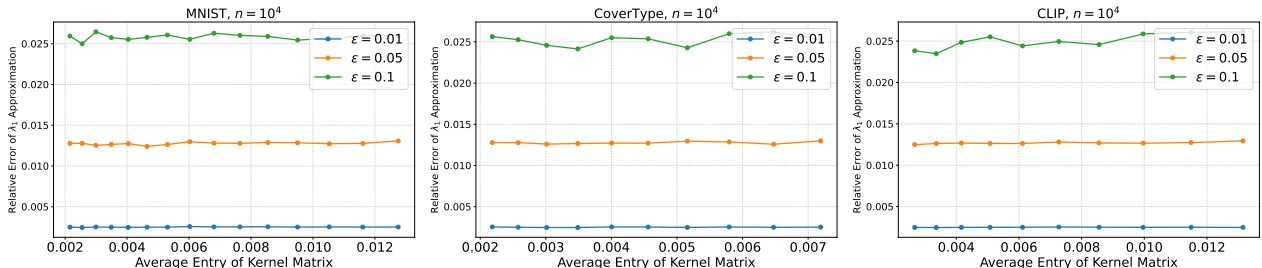

*Figure 9.* We vary the average kernel matrix entry (by varying the scale $\sigma$), and plot the relative error of computing $\lambda_1$ using 75 power method iterations, across three choices of $\varepsilon$.

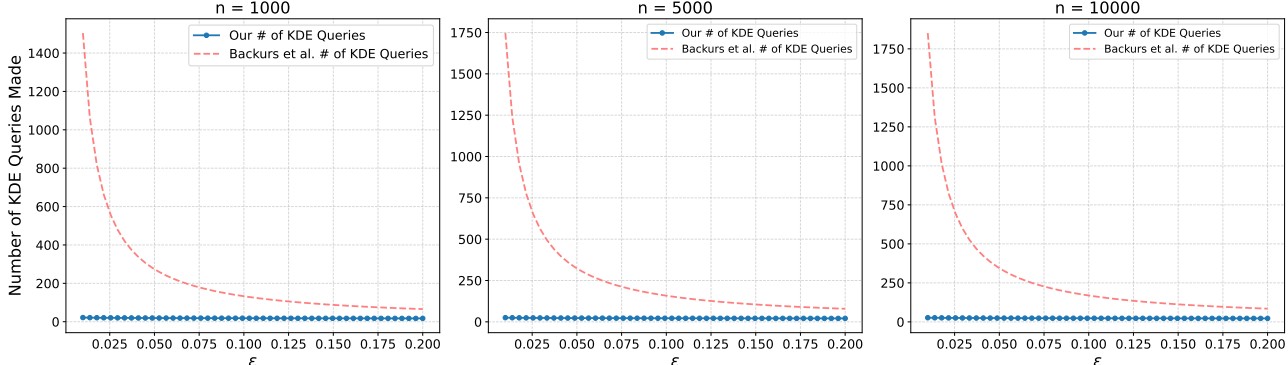

*Figure 10.* Comparing the number of KDE queries made.

### G.5. Comparing Our Gaussian MVP Computational Cost to That of (Backurs et al., 2021)

Lastly, we demonstrate the efficiency of our MVP approach for the Gaussian kernel (see Theorem 2.1) versus that of (Backurs et al., 2021), by comparing their running times in practice. First, we note that algorithms (e.g. the ones in Table 3) that answer KDE queries have an *inverse polynomial* dependence of $\mu$, the additive error. This means that the smaller additive error we require, the slower the KDE query is, as expected.

Now we note that in our algorithm for Theorem 2.1, we always use a *much larger* value of $\mu$ than the corresponding algorithm of (Backurs et al., 2021): our value is at least $1/\varepsilon$ times larger. Furthermore, the runtimes of both our algorithm and that of (Backurs et al., 2021) is dominated by KDE calls. This means that to demonstrate that our algorithm is more computationally efficient, it suffices to compare the total number of KDE queries made by our algorithm and theirs (in fact such a comparison even gives an advantage to (Backurs et al., 2021) since we use larger $\mu$ values). Thus, we plot the exact number of KDE queried used by both algorithms for different values of $n$ as a function of $\varepsilon$. For both algorithms, this is exactly equal to the number of buckets or partitions of the coordinates of the query vector that the algorithms initialize, since both methods perform one KDE call per partition.

Fig. 10 illustrates the number of queries for our method (given by roughly $\log_2(n^{1.5}/\epsilon)$, shown in blue with markers), versus the number of queries required by (Backurs et al., 2021) (given by roughly $\frac{\log_2(n^{1.5}/\epsilon)}{\log_2(1+\epsilon)}$, shown in dashed red). As evident from the plot, our method requires significantly fewer queries. For example, with $n = 1000$ and $\epsilon = 0.1$, our method performs approximately 7**x fewer queries**. Indeed, we observed at least this fraction reduction in the running time when we implemented our versus their 0.1-approximate MVP, using the optimized random sampling as described before for the underlying KDE method.

Thus, no matter what the underlying KDE query implementation is, either the theoretically SOTA in Table 3 or a heuristic implementation, for realistic values if $n$ and $\varepsilon$, we use a substantially fewer number of KDE queries (and each of our KDE queries is less costly as well!).

