# OpenReview forum: "Even Faster Kernel Matrix Linear Algebra via Density Estimation"
_ICML.cc/2026/Conference — ICML 2026 spotlight_

### Official Review · Reviewer_4K6z · 2026-03-06

**Soundness:** 4
**Presentation:** 3
**Significance:** 3
**Originality:** 3
**Overall Recommendation:** 4
**Confidence:** 3

**Summary:**

The authors give a number of upper and lower bounds for various linear algebra problems involving kernel matrices associated with the gaussian and laplacian kernel functions. They also present empirical evidence that demonstrates their algorithms’ superior performance.

 With the use of black box KDEs, they shave off a factor of 1/(epsilon^(1+p)) in the running time for non-negative (kernel) approximate matrix-vector multiplication by employing a coarser, adaptive bucketing scheme, which reduces the number of KDE calls required.

Using their approximate M-v algorithm as a building block, this lets them improve on the previous state of the art for approximating the top eigenvalue of gaussian/laplacian kernel matrices.

They also give an algorithm for approximating kernel sums with a more modest improvement, along with various lower bounds. In particular, they show that approximating gaussian kernel quadratic forms up to polynomial relative error requires almost quadratic time, assuming SETH.

**Compliance With Llm Reviewing Policy:**

Affirmed.

**Key Questions For Authors:**

How do we extend Algorithm 1 and 2 to kernel functions with parameterised bandwidths? How does this affect the complexities?

The Backurs et al., 2021 algorithm is an eps-non-negative MVP algorithm with running time $…/\epsilon^{(3+2p)}$ (line 598) and Theorem A.3 suggests using it as a black box for Algorithm 2 should result in a running time of $…/(\epsilon^{(4+2p)}$. I guess this implies that their running time for Top eigenvalue can be reduced from $…/\epsilon^{(7.692)}$ to $…/\epsilon^{(4+2p)}=.../\epsilon^{(4.346)}$ in Table 1 for the gaussian kernel?

Should the t in $\mathcal{D}_t([x_i ; 0])$ (Line 12 of Algorithm 1) actually be the transformed version of $\mathcal{X}_t$? From Definition A.1, I expected the subscript of $\mathcal{D}$ to be a set.

In Lemma B.3, the subscript $\ell$ is confusing me because it is used inside and outside the sum (line 737 for example). Is there a clearer way of writing this or am I missing something obvious?

**Limitations:**

Yes

**Strengths And Weaknesses:**

The M-v and top eigenvalue algorithms appear to be strong contributions and the experiments highlight this clearly. Theorem 2.7 and (its corollaries) appears to be the most relevant lower bound. I don’t think many readers will find Theorem 2.6 interesting due to its artificial nature.

Maybe I missed it, but I didn’t see how to extend the algorithms to kernel functions with non-unitary bandwidths. In the experiments, it’s clear that this is possible. Stating clearly how to do this would strengthen the significance of this paper as it would help those who actually want to implement the algorithms themselves.

---

> ### Author Rebuttal · Authors · 2026-03-25
>
> Thank you very much to the reviewer for their feedback! We respond to their concerns and comments below.
>
> > "I don’t think many readers will find Theorem 2.6 interesting due to its artificial nature."
>
> While the lower bound problem is slightly different, we view our lower bound of Theorem 2.6 as taking the first preliminary step towards a full theoretical understanding of the complexity of kernel MVPs. We note that several works have conjectured the necessity of quadratic time for obtaining a relative error approximation for computing $Ky$ for general vectors $y$ (e.g. [1]), but so far a theoretical proof has not been established.
>
> [1] Indyk, P., Kapralov, M., Sheth, K., and Wagner, T. Improved algorithms for kernel matrix-vector multiplication under sparsity assumptions. ICLR 2025.
>
>
> > "How do we extend Algorithm 1 and 2 to kernel functions with parameterised bandwidths? How does this affect the complexities?
>
> Our algorithms work with the same complexity and approximation guarantees for arbitrary bandwidth by simply scaling the points before performing any computation. For example in the Gaussian kernel case, suppose we have data points $x_1, \cdots, x_n$ and bandwidth $\sigma^2$. Then if we consider the data points $x_i’ = x_i / \sigma$, we have that $e^{- \|x_i - x_j\|_2^2/\sigma^2} = e^{- \|x_i’ - x_j’\|_2^2}$. Thus, we can always assume the bandwidth is 1 by scaling, and a similar reasoning works for all the kernels we consider in the paper (Gaussian, Exponential, Laplacian, Rational Quadratic).
>
> > "The Backurs et al., 2021 algorithm is an eps-non-negative MVP algorithm with running time (line 598) ... I guess this implies that their running time for Top eigenvalue can be reduced in Table 1 for the gaussian kernel?
>
> Yes, that is correct in principle, but only if one optimizes the proof of Backurs et al., 2021. However a completely different type of analysis is required to get our best bound, which we discuss in Section 2.2.
>
> > "Should the t in (Line 12 of Algorithm 1) actually be the transformed version of X_t?"
>
> Yes that is correct. We will update the text to be more clear.
>
> > "In Lemma B.3, the subscript is confusing me because it is used inside and outside the sum (line 737 for example). Is there a clearer way of writing this or am I missing something obvious?"
>
> Thank you for catching that. We will change the subscript notation for the outside variable to be more clear.

---

> > ### Author Rebuttal · Reviewer_4K6z · 2026-04-01
> >
> > Thank you for your clear rebuttal. My concerns have been adequately addressed.

---

### Official Review · Reviewer_tvPu · 2026-03-11

**Soundness:** 3
**Presentation:** 3
**Significance:** 3
**Originality:** 3
**Overall Recommendation:** 5
**Confidence:** 3

**Summary:**

This paper investigates the application of kernel density estimation (KDE) queries to several fundamental linear algebra approximation tasks (matrix-vector products, matrix-matrix products, spectral norm, and matrix sum) involving the kernel matrix and improves upon the best existing $(1\pm\epsilon)$-approximate algorithms in terms of time complexity, significantly reducing the dependency on the data size $n$ and the approximation error $\epsilon$. The work makes a significant theoretical contribution to downstream approximation algorithms for kernel matrix linear algebra and advances the broader application of kernel methods in machine learning.

**Compliance With Llm Reviewing Policy:**

Affirmed.

**Key Questions For Authors:**

See "Strengths And Weaknesses".

**Limitations:**

Yes.

**Strengths And Weaknesses:**

**Strengths**

1.The paper has made substantial progress on several approximate linear algebra problems for kernel matrices. For example, for the Non-negative Matrix-Vector Product problem, it reduced the time complexity of the best-known result by a factor of approximately $\frac{1}{\varepsilon^{1.173}}$​. For the Top Eigenvalue estimation problem, it achieved a reduction by a factor of approximately $\frac{1}{\varepsilon^{4.519}}$​. For the Kernel Sum problem, the improvement is by a factor of approximately $\frac{n^{0.07}}{\varepsilon^{0.16}}$​.

2.The paper not only improves the upper bounds but also theoretically proves quadratic lower bounds on the time complexity for computing the aforementioned kernel linear algebra problems. In particular, the work demonstrates that in the asymmetric case, computing the matrix sum, the top singular value, and non-negative matrix-vector products for the Gaussian kernel—even up to polynomial approximation factors—solves the Orthogonal Vectors (OV) problem and therefore requires nearly quadratic time.

**Weaknesses**

1.In the paper, the core premise of all upper bounds (algorithms) is the existence of an efficient KDE data structure satisfying Definition A.1, and the algorithm's complexity depends on the key parameter $p$ of this data structure. For kernels not listed in the paper, would a larger $p$ value significantly increase the time complexity?

2.The paper's core algorithm (e.g., the matrix-vector product in Theorem A.1) relies on the assumption that the input vector is non-negative. For general vectors, the paper considers the problem of computing matrix-vector products for a matrix $K'$ in Theorem 2.6, providing a conditional (based on SETH) quadratic-time complexity result. Does this complexity completely represent the original problem? Do existing works discuss whether the non-negativity condition is "necessary" for achieving relative-error approximate matrix-vector products?

---

> ### Author Rebuttal · Authors · 2026-03-25
>
> Thank you very much to the reviewer for their feedback! We respond to their concerns and comments below.
>
> > "In the paper, the core premise of all upper bounds (algorithms) is the existence of an efficient KDE data structure satisfying Definition A.1, and the algorithm's complexity depends on the key parameter of this data structure. For kernels not listed in the paper, would a larger value significantly increase the time complexity?"
>
> To the best of our knowledge, we have listed all the kernels that we are aware of having KDE data structures with p < 1 (Gaussian, Exponential, Laplacian, Rational Quadratic). For other kernels (or any bounded function), it can be easily shown that a KDE data structure always exists with p = 1 by just uniform sampling. In general, our bounds abstract away the KDE data structure so it does not rely on any particular instantiation of a KDE data structure. This means that faster KDE data structures (even if they use very different underlying techniques)  automatically imply faster downstream algorithm via our work.
>
> > "The paper's core algorithm (e.g., the matrix-vector product in Theorem A.1) relies on the assumption that the input vector is non-negative. For general vectors, the paper considers the problem of computing matrix-vector products for a matrix in Theorem 2.6, providing a conditional (based on SETH) quadratic-time complexity result. Does this complexity completely represent the original problem? Do existing works discuss whether the non-negativity condition is "necessary" for achieving relative-error approximate matrix-vector products?"
>
> The complexity is for a different, but related problem where the kernel matrix’s rows and columns are allowed to be indexed by different point sets, but only up to one point different. Furthermore, we change the kernel to be $e^{-\|x+y\|_2^2}$. Please see our response to Reviewer vde2 why our techniques require this.
>
> There are several works that **conjecture** the necessity of quadratic time for obtaining a relative error approximation for computing $Ky$ for general vectors $y$ (e.g. [1]), but so far a theoretical proof has not been established. We view our lower bound of Theorem 2.6 as taking the first preliminary step towards a full theoretical understanding of the complexity of kernel MVPs.
>
> [1] Indyk, P., Kapralov, M., Sheth, K., and Wagner, T. Improved algorithms for kernel matrix-vector multiplication under sparsity assumptions. ICLR 2025.

---

> > ### Author Rebuttal · Reviewer_tvPu · 2026-04-03
> >
> > I believe the author addressed my issue well, and I will maintain my positive score.

---

### Official Review · Reviewer_vde2 · 2026-03-12

**Soundness:** 4
**Presentation:** 4
**Significance:** 3
**Originality:** 3
**Overall Recommendation:** 5
**Confidence:** 4

**Summary:**

The authors study applications of kernel density estimation (KDE) to various linear algebraic tasks for the kernel matrix of a collection of $n$ data points in $\mathbb{R}^d$.

In the classical kernel density estimation problem (specialized here to the Gaussian kernel), the input is a set of $n$ points $x_1, \ldots, x_n \in \mathbb{R}^d$ (denoted later by $X$) and error parameters $\varepsilon, \mu \in (0,1)$. The task of the algorithm is to construct a data structure $\mathcal{D}$ such that for any query $y \in \mathbb{R}^d$, the estimate $\mathcal{D}(y)$ satisfies

$ \sum_{i=1}^n \frac{\exp(-\lVert y - x_i\rVert_2^2)}{n}
\le
\mathcal{D}(y)
\le
(1+\varepsilon)\sum_{i=1}^n \frac{\exp(-\lVert y - x_i\rVert_2^2)}{n}+ \mu .
$

This data structure can be constructed in time $\tilde{O}(dn/\mu^p)$ for $p = 0.173 + o(1)$ by previous work of Charikar et al. (2020).

The authors utilize connections between various linear algebraic problems and KDE to significantly improve the runtime of several tasks: approximate matrix-vector products (given non-negative $y$ and precision $\varepsilon$, return $x = Ky + e$ such that $\lVert e \rVert_2 \le \varepsilon \lVert Ky \rVert_2$), approximating the top eigenvalue of a kernel matrix, and non-negative vector-matrix-vector products (computing a $(1+\varepsilon)$ approximation to $y^{\top}Ky$).

In particular, previous approaches for estimating approximate matrix-vector products rely on the following insight. To estimate the first coordinate
\[
(Ky)_1 = \sum_{j=1}^n y_j k(x_1, x_j),
\]
we observe that this quantity is very close to a KDE query but weighted by coefficients $y_j$. To handle this, (1) we ignore all $y_j \le \varepsilon / n^{1.5}$ since these values contribute very little to the overall sum, and (2) we bucket the remaining coordinates into geometrically increasing ranges by a factor of $(1+\varepsilon)$. Since the values of the coordinates are roughly the same within each bucket (up to a $(1+\varepsilon)$ factor), this value can be factored out of the sum in each bucket. Thus, the problem reduces to a standard KDE query within each of the $O(\log(n/\varepsilon)/\varepsilon)$ buckets.

The main contribution of the new algorithm is to show that the number of buckets can be reduced to $O(\log(n / \varepsilon))$ by instead splitting into buckets that grow by a factor of $2$ (which introduces several technical challenges), and then showing how to reduce the problem to a single KDE query. This algorithm is then used as a subroutine for estimating the top eigenvalue of $K$ (by adapting the power method) and for vector-matrix-vector queries (by computing the inner product between $y$ and $Ky$).

**Compliance With Llm Reviewing Policy:**

Affirmed.

**Final Justification:**

The author's response was very helpful and addressed all of my questions. I believe this is a fundamental problem for both theory and practice. Overall, I recommend accepting the paper to ICML.

**Key Questions For Authors:**

- Are there any known lower bounds for estimating $Ky$ when $y$ has non-negative entries?
- Can you provide some more intuition on why you needed to prove the conditional lower bound for a different kernel function (with a plus)? What are the technical barriers for extending the conditional lower bound to other Kernels?

**Limitations:**

Yes.

**Strengths And Weaknesses:**

Strengths:
- The problem is well motivated, and the new algorithms significantly improve the sample complexity over previous algorithms in the literature. All claims are well-supported by rigorous proofs.
- The paper is very well structured and arguments are clearly explained. The technical overview does a good job of introducing previous techniques and describing the new ideas which were needed to surpass the previous best runtime.
- Positive results which may appear restrictive (i.e. matrix-vector products for non-negative y) are accompanied with conditional lower bounds (assuming SETH).
- I appreciate that the authors gave a comprehensive discussion of open problems stemming from their work.

Weaknesses:
- Algorithms are for Laplace and Gaussian kernels only.
- The conditional lower bound for matrix-vector products is for a slightly different (but related) problem.

---

> ### Author Rebuttal · Authors · 2026-03-25
>
> Thank you very much to the reviewer for their feedback! We respond to their concerns and comments below.
>
> > "Algorithms are for Laplace and Gaussian kernels only."
>
> We would like to point out that our techniques also extend to the Exponential and Rational quadratic kernels. While our faster approx MVP algorithm only holds the Laplace and Gaussian kernels, our improved power method analysis gives the state of the art results for approximation the top eigenvalue of the Gaussian, Laplace, Exponential, and Rational Quadratic kernel. Furthermore, we also obtain the state of the art result for all Laplace, Gaussian, and Exponential  kernels for approximating the sum of the kernel matrix up to $1+\epsilon$ multiplicative error. Please see Appendix A for details.
>
> > "Are there any known lower bounds for estimating Ky when y has non-negative entries?"
>
> There are no lower bounds currently known, besides the trivial $\Omega(n)$  time required to read all the entries. However, we give the first evidence that a running time of the form $n^{1+p}$, where $p$ is the exponent in the Gaussian KDE, maybe necessary for an approximate MVP algorithm for $Ky$ even when $y$ has non-negative entries: in Lemma B5 we show how any non-negative MVP algorithm with similar guarantees as our upper bound can be used to answer n different KDE queries. Thus, this presents evidence that the running time estimating $Ky$ should scale with $n$ times the running time of answering a single KDE query, which is what our upper bound achieves.
>
> >"Can you provide some more intuition on why you needed to prove the conditional lower bound for a different kernel function (with a plus)? What are the technical barriers for extending the conditional lower bound to other Kernels?"
>
> The main technical barrier is that we are trying to embed the orthogonal vectors problem (OV) in computing Ky for a y with possibly negative entries. This is a popular lower bound hypothesis that conjectures the following problem needs quadratic time: given two sets $A, B$ each of $n$ binary vectors, find a pair that $a \in A, b \in B$ such that $\langle a,b\rangle = 0$. By padding the vectors, we can also assume that the number of 1’s in the vectors is the same value $t$.
>
> Now $\|a+b\|^2 = 2t + 2\langle a,b\rangle$ so $e^{- C \cdot \| a+b \| ^2} = e^{-2Ct} \cdot  e^{- 2C \langle a,b\rangle}$. Thus, the entries of the matrix that correspond to an orthogonal pair have value $e^{-2Ct}$, whereas entries that correspond to **non** orthogonal pairs are much smaller, since we multiply by a value $e^{- 2C \langle a,b\rangle} \ll 1$ for a sufficiently large $C \gg 1$. Thus, our choice of a function **boosts** the signal that there is an orthogonal pair, whereas for the other choice of  $e^{- C \cdot \| a - b \|^2}$, the signal of an orthogonal pair is actually **diminished**, which is the technical barrier.

---

> > ### Author Rebuttal · Reviewer_vde2 · 2026-04-02
> >
> > Thank you for your response and for addressing my questions! I will keep my positive score as is.

---

### Official Review · Reviewer_wYdf · 2026-03-12

**Soundness:** 3
**Presentation:** 3
**Significance:** 3
**Originality:** 3
**Overall Recommendation:** 4
**Confidence:** 4

**Summary:**

The paper presents a new class of algorithms for conducting efficient operations on kernel matrices for Gaussian kernels K. Two main applications are: fast (sub-quadratic) approximate matrix-vector multiplication with matrices K and fast (sub-cubic) approximate estimation of the top eigenvalues of K. Efficient algorithms solving these problems find applications in several fields of machine learning such as: standard kernel-based algorithms, interpolation methods with kernels, 3D modeling, and more.

**Compliance With Llm Reviewing Policy:**

Affirmed.

**Final Justification:**

I will keep my original positive score. The empirical evaluation for the presented algorithm is very limited, but the method is important and impactful in many fields of machine learning.

**Key Questions For Authors:**

The reviewer would like the Authors to comment on the lack of the empirical tests of the presented methods and ask them to add some in the rebuttal period is feasible (these do not need to be large-scale ML experiments).
The reviewer would like to also ask the Authors whether presented techniques can be extended beyond Gaussian kernels.

**Limitations:**

yes

**Strengths And Weaknesses:**

Strengths:

The paper provides novel theoretical contributions. Main theoretical results: Theorem 2.1 and Theorem 2.2 (with the generalization, namely Theorem A.3 provided in the Appendix) are important contributions to the field. They improve upon the already strong results and even though the removal of the polynomials of the (1 / epsilon) factors might seem incremental, it is of great practical importance and thus should be of great interest to the ML Community. The theoretical work is very clearly presented, with all details given.

Weaknesses:

The paper does not have an empirical evaluation of the presented methods. It goes without saying that the main theoretical contribution is already nontrivial and important for the Community, though some empirical evidence, especially given widespread potential downstream applications of the presented mechanisms, would naturally complement strong theoretical analysis. A natural baseline would be standard kernel algorithms, e.g. kernel ridge regression.

---

> ### Author Rebuttal · Authors · 2026-03-25
>
> Thank you very much to the reviewer for their feedback! We respond to their concerns and comments below.
>
> > "The reviewer would like the Authors to comment on the lack of the empirical tests of the presented methods and ask them to add some in the rebuttal period is feasible (these do not need to be large-scale ML experiments)."
>
> We would like to point out that **we have already included empirical evaluations** in the submission; please see Appendix G where we perform some empirical evaluations for calculating the top eigenvalue and approximate matrix vector products.
>
> > "The reviewer would like to also ask the Authors whether presented techniques can be extended beyond Gaussian kernels."
>
> Our theoretical results extend to Exponential, Laplacian, and the rational quadratic kernels as well. Please see Appendix A where we discuss extensions to these kernels.

---

> > ### Author Rebuttal · Reviewer_wYdf · 2026-04-03
> >
> > My concerns have been addresses. Thus I will remain my positive score.

---

> > > ### Author Response · Authors · 2026-04-03
> > >
> > > Thank you for the feedback! Given that your main weaknesses were the lack of empirical evaluations and extension to other kernels (both of which were already included and addressed in the original submission), we are wondering if you would consider adjusting your scores.
> > >
> > > Many thanks,
> > > The authors

---

### Decision · Program_Chairs · 2026-04-30

**Decision:**

Accept (spotlight)

**Comment:**

The paper presents improved algorithms for computing matrix-vector products, and related products for kernel matrices. All the reviewers appeciate the significance of the work, and gave positive evaluation. Given the results and the importance of the studied problems in machine learning community, I recommend accept.